# The hysteretic response of a shallow pyroclastic deposit

Luca Comegna[1], Emilia Damiano[1], Roberto Greco[1], Lucio Olivares[1], Luciano Picarelli[2]

[1]Department of Engineering, University of Campania "Luigi Vanvitelli", Aversa, 81031, Italy
[2]JTC1 "Natural Slopes and Landslides", Chair, Federation of International Geo-Engineering Societies (FedIGS), Naples, 80131, Italy

*Correspondence to*: Luca Comegna (luca.comegna@unicampania.it)

**Abstract.** In the last decades, in Campania (Southern Italy), steep slopes mantled by loose air-fall pyroclastic soils have been the seat of shallow fast rainfall-induced landslides. The occurrence of such events has been the result of the combination of critical rainstorms and of unfavourable initial conditions determined by antecedent infiltration/evaporation/drainage processes. In order to understand the nature of the phenomena at hand and to clarify the role of all influencing factors, an automatic monitoring station has been installed in an area already subject to a recent killer flowslide (December, 1999). The paper reports data collected in 2011 about volumetric water content and suction (used to investigate the soil water retention features) and rainfall depth and temperature (providing the boundary conditions). In particular, the installation at the same depths of tensiometers and Time Domain Reflectometry (TDR) sensors allowed to recognize the hysteretic nature of the wetting/drying soil response to weather forcing and its influence on the slope stability conditions.

The data reported in the paper are freely available at https://doi.org/10.5281/zenodo.4281166 (Comegna et al., 2020).

## 1 Introduction

The hydraulic response of unsaturated soils subjected to infiltration and/or evaporation phenomena is usually modelled through the well-known *Soil Water Retention Curve*, SWRC, correlating matric suction, $s$, with volumetric water content, $\theta$. Experimental evidence and theoretical considerations (e.g., Mualem, 1976; Pham, 2002; Wheeler et al., 2003; Tami et al., 2004; Li, 2005; Tarantino, 2009; Yang et al., 2012; Pirone et al., 2014; Comegna et al., 2016c; Chen et al., 2017; Chen et al., 2019; Rianna et al., 2019) indicate that the SWRC is not univocal, but may depend on soil initial conditions and on the induced wetting or drying paths. This soil response, known as *hydraulic hysteresis*, may be related to microscopic phenomena affecting the energy state of water at pore scale (i.e. variations of contact angle during solid particles wetting and drying, or bottlenecks differently affecting filling and emptying of pores), as well as macroscopic phenomena depending on the boundary conditions and on the rate of the specific transient wetting/drying process (e.g. air entrapment). Figure 1 shows the typical response of an initially saturated soil sample subjected to drying and wetting cycles. During the first drying stage, $\theta$ decreases from the initial value, $\theta_{s,d}$, following a path, known as the *main drying curve*, until attaining the minimum, corresponding to a high $s$ value, known as the "residual volumetric water content", $\theta_r$. In the subsequent wetting process, $\theta$ increases along a different path, known as the *main wetting curve* (Fig. 1), until reaching a final maximum value, $\theta_{s,w}$, at $s = 0$: $\theta_{s,w}$ is usually different from

$\theta_{s,d}$ because of some air entrapment that does not allow full soil saturation. However, in some cases, if the wetting process is very slow, it may occur that $\theta_{s,w} \cong \theta_{s,d}$.

If a reverse process takes place along one of these paths, the main path is abandoned and a different *scanning curve*, located between the two main paths, is then travelled (Fig. 1). Scanning curves, which in turn may be characterised by internal hysteretic loops, present a lower slope than the main curves: this physically means that, starting from the same $s$ value, the $\theta$ variation corresponding to a given $s$ change is smaller running a scanning than a main curve. As shown in Figure 1, the final part of a scanning path may coincide with the nearest primary curve. This result has been experimentally recognized by Tami et al. (2004) through some tests carried out on a 30° model slope consisting of a 40 cm thick layer of fine sands, overlying a 20 cm thick layer of gravelly sands, subjected to artificial precipitation of different intensity. Figure 2a shows the scanning curves obtained by fitting the coupled data measured by a tensiometer and a Time Domain Reflectometry (TDR) probe located at a depth of 30 cm (Fig. 2b) during two consecutive drying stages (1-2, 2-3) and two consecutive wetting stages (4-5, 5-6). The main drying and wetting curves had been independently obtained through a Tempe cell and capillary rise tests. Therefore, the same $\theta$ may be associated with different water potential energies, thus with different $s$ values within an interval defined by the highest and lowest limits respectively imposed by the main drying curve and the main wetting curve.

Even though often observed in laboratory experiments, the hydraulic hysteretic response of unsaturated soils is still often neglected at slope scale, being usually modelled by a single SWRC fitting all the available experimental data. This choice is frequently due to unavailability of detailed field information. Most of the knowledge is in fact based on the results of laboratory investigations and/or of physical modelling. These tests, although very useful, are unavoidably not able to take into account further aspects that could make the actual hysteretic response of natural slopes quite different from what is observed in the lab, as the influence of different boundary conditions, the role of root water uptake or of the atmospheric evaporative demand, etc. Considering that the hydrological response could affect the stability conditions of natural slopes, an automatic monitoring station was installed in a shallow layer of loose pyroclastic soils covering a steep mountainous area in a site of Campania Region (Southern Italy), which in 1999 was the seat of a rainfall-induced flowslide (Damiano et al., 2012). The availability of continuous data, consisting of rainfall depth, temperature, volumetric water content and suction readings, allowed to collect useful information from January, 2011, to January, 2012, that have been also used to estimate the slope stability conditions at the investigated depths.

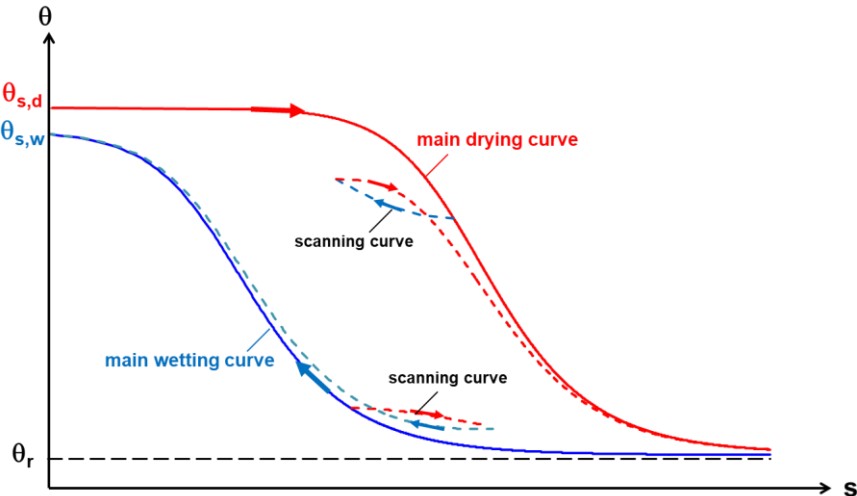

**Figure 1: Typical hydraulic response of a soil sample subjected to drying and wetting cycles. Along the *main drying curve*, the volumetric water content, $\theta$, continuously decreases with the matric suction, $s$, from the initial saturated value, $\theta_{s,d}$, until the minimum residual value, $\theta_r$, is reached. Along the *main wetting curve*, $\theta$ continuously increases from the initial residual value, $\theta_r$, until the final maximum value, $\theta_{s,w}$, is reached at $s = 0$. If the continuous main drying (wetting) stage is interrupted by a reverse wetting (drying) process, a different *scanning curve*, located under (over) the main path, is travelled.**

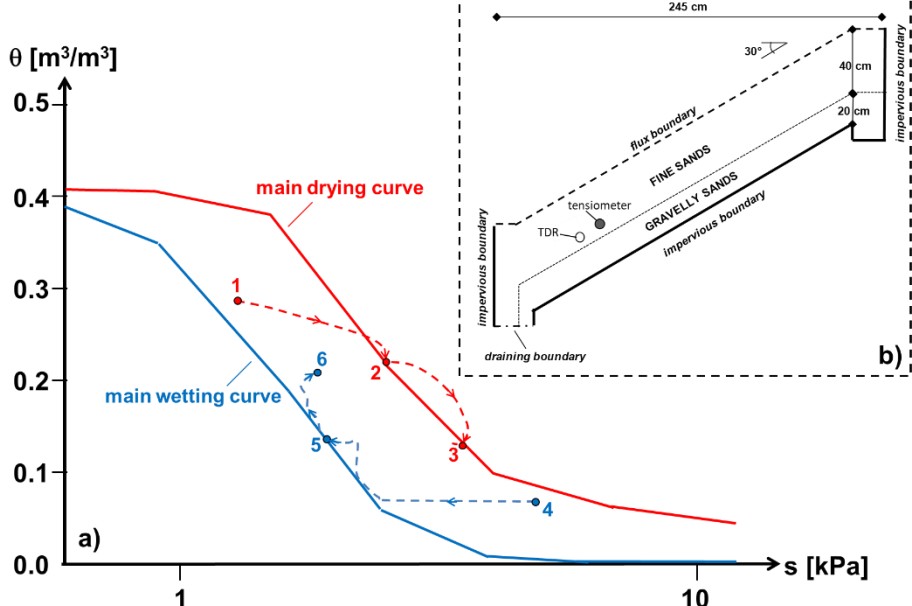

**Figure 2: Scanning curves (a) obtained from flume tests carried out by Tami et al. (2004) during two consecutive drying stages (1-2, 2-3) and wetting stages (4-5, 5-6). The 30° inclined model slope (b) consists of a 40 cm thick layer of fine sands that overlies a 20 cm thick layer of gravelly sands. The volumetric water content, $\theta$, and the matric suction, $s$, are respectively measured by a Time**
**Domain Reflectometry (TDR) probe and a tensiometer, both installed at a depth of 30 cm. The main drying curve and the main wetting curve have been obtained through a Tempe cell and capillary rise tests.**

## 2 Data and methods

### 2.1 Geomorphological and climate framework

The investigated site is located at an elevation of 560 m a.s.l., on the North-East facing slope of mount Cornito (Fig. 3a), 2 km from the town of Cervinara (Campania Region, Southern Italy), about 50 km northeast of Naples. On December, 16[th], 1999, the slope was involved in a disastrous flowslide induced by a rainstorm of 320 mm in 50 hours. The landslide body entered the narrow Cornito stream reaching the downslope town (Fig. 3b), where it caused heavy damage and five human deaths. Geological surveys and geotechnical investigations reveal that the basal Mesozoic–Cenozoic fractured limestones are overlain by air-fall sandy soils resulting from the explosive volcanic activity of Somma–Vesuvius and Phleagraean Fields (e.g., Fiorillo et al., 2001; Damiano et al., 2012). In particular, the pyroclastic deposits consist of alternating layers of ashes and pumices, more or less parallel to the bedrock surface, with a thickness strongly dependent on the slope angle, ranging from some decimetres in the steepest upslope zones (about 50° inclined) to more than 10 m at the foot of the hill (Guadagno et al., 2011). In some verticals, some layers were not found, possibly as a result of past landslides or of erosive processes.

Cultivated chestnut trees are widespread on the slope, except some areas where the vegetation consists of shrubs and grass. When the tree foliage is present, usually from May to late September, a dense underbrush grows, mainly formed by ferns and other seasonal shrubs. Differently, in October leaves fall from the trees, and the underbrush disappears until the following late spring. During late autumn and winter the ground is mostly covered by a layer of litter, mainly originating from fallen chestnut leaves. The seasonal variations of vegetation affect the soil hydrologic response to meteorological forcing by: i) interception of the precipitation and ii) root water uptake (Comegna et al., 2013). Interception is caused by canopy, understory and litter. The total evapotranspiration flux, distributed over the root depth according to the local value of soil water potential, is highly variable throughout the year owing to the dormant leafless vegetation in winter. Visual inspections in trenches dug during the investigations that have been carried out on site showed that roots usually extend across the entire soil depth up the basal limestones, with a maximum density within the uppermost 0.50 m of soil cover, becoming sparse below the depth of 1.50 m.

Concerning climate, Figure 4 shows mean monthly values of rainfall depth. These have been calculated with the 2001-2017 data from the rain gauge installed in Cervinara, at the elevation of 320 m a.s.l., by the meteorological alert network managed by the "Functional Centre for forecast, prevention and monitoring of risks and alerting for civil protection" of Campania Civil Protection Agency. The mean annual rainfall depth is around 1600 mm: as typical of the Mediterranean climate, most precipitation occur between October and April, while summer is essentially dry. Figure 4 also reports mean monthly values of temperature. Those have been calculated with the data monitored from 1979 to 1998 by the meteorological station of Montesarchio, managed by the National Hydrological Service. This station is located 4 km from the test site and approximately at the same elevation. These data have been used for the estimation with the Thornthwaite expression (1946) of the monthly potential evapotranspiration, PET. In particular, the estimated PET annual value is around 750 mm (Marino et al., 2020).

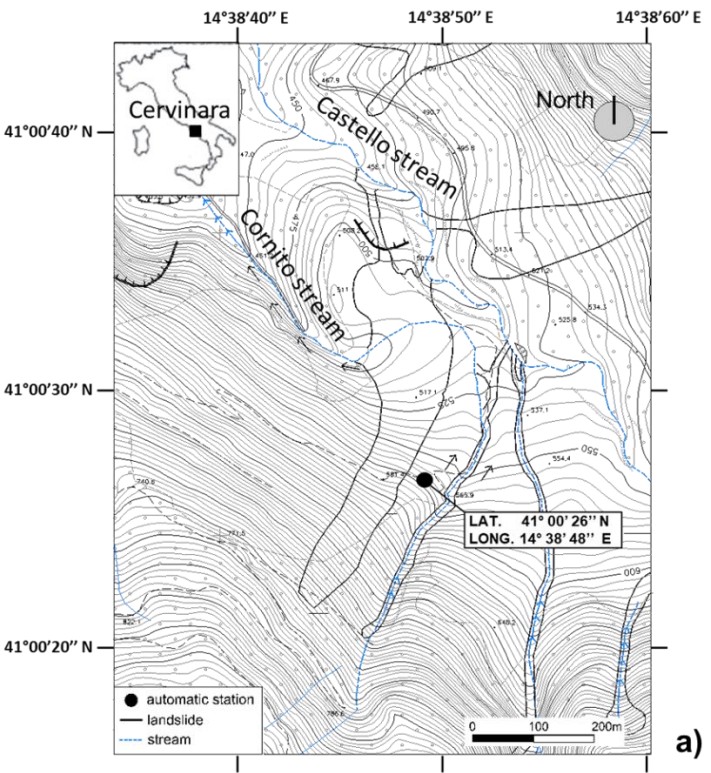

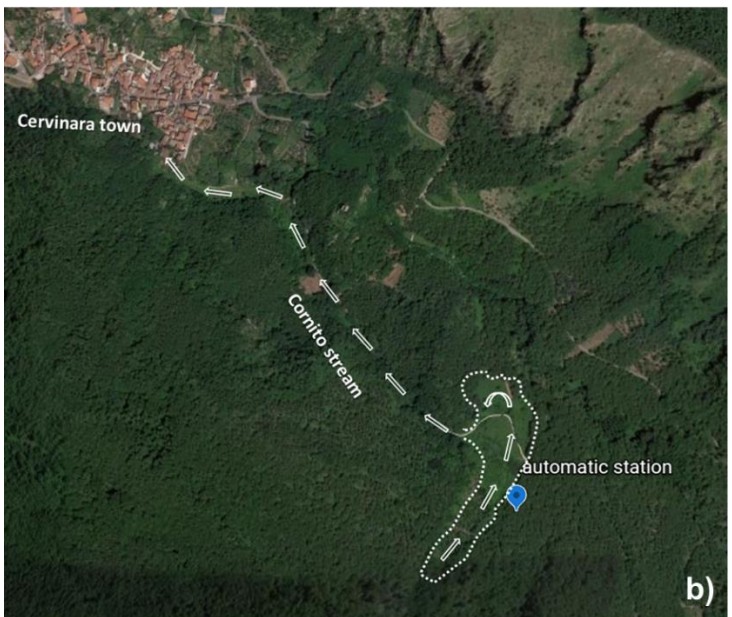

Figure 3: Location of the automatic monitoring station installed on the North-East facing slope of mount Cornito, 2 Km from the Cervinara town (Campania Region, Southern Italy): topographic map (a) and aerial photo of the site (b). On December, 16th, 1999, the slope was involved in a flowslide that entered the narrow Cornito stream reaching the downslope town. Map (a) is provided by Damiano (2004). Aerial photo (b) is provided by © Google Earth 2020.

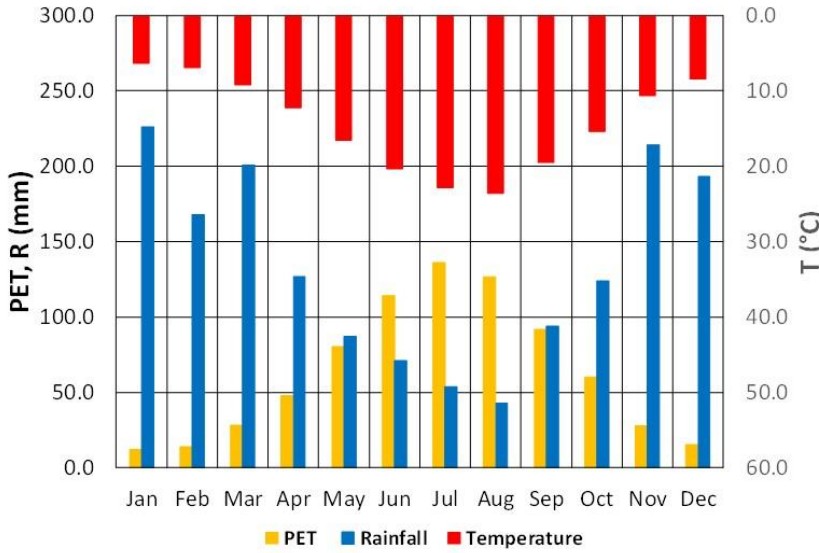


**Figure 4: Mean monthly values of rainfall depth, R, temperature, T, and potential evapotranspiration, PET. R is calculated with the 2001-2017 data from the rain gauge installed in Cervinara by the meteorological alert network managed by the "Functional Centre for forecast, prevention and monitoring of risks and alerting for civil protection" of Campania Civil Protection Agency. T is calculated with the data monitored from 1979 to 1998 by the meteorological station of Montesarchio, managed by the National**
**Hydrological Service. PET is estimated with the Thornthwaite expression (1946).**

## 2.2 Soil properties

Thanks to a number of field investigations and geotechnical laboratory tests (carried out both on undisturbed and reconstituted soil samples), Damiano et al. (2012) provide information about the physical and mechanical soil properties (Table 1). In the monitored verticals, the soil deposit is 1.9 m deep with a sloping angle of about 40° (Fig. 5). The local stratigraphy consists of
the following unsaturated soil layers: 1) topsoil, 10 cm thick; 2) coarse pumices, 40 cm thick; 3) ashes, 1.30 m thick; 4) altered ashes, 10 cm thick, located just above the bedrock. The volcanic ash is a sandy silt, the pumices are sandy gravels. The lowermost altered ashes overlying the bedrock are representative of a deteriorated thin ash layer, with a grain size which is turning from sandy silt to clayey and silty sand, featured by a plasticity index ranging in the interval 10–30%. The soil porosity ranges between 50% and 75%. The shear strength parameters are typical of essentially cohesionless coarse grains soils.


**Table 1: Main physical and mechanical properties of the pyroclastic cover: specific unit weight, $\gamma_s$; unit weight, $\gamma$; porosity; cohesion, c'; friction angle, $\varphi$'.**

| layer | $\gamma_s$ [kN/m³] | $\gamma$ [kN/m³] | porosity [%] | c'[kPa] | $\varphi$'[°] |
|---|---|---|---|---|---|
| coarse pumices | 23 | 13 | 50-55 | 0 | - |
| ashes | 26 | 14 | 68-75 | 0 | 38 |
| altered ashes | 26 | 16 | 60 | 2 | 38 |

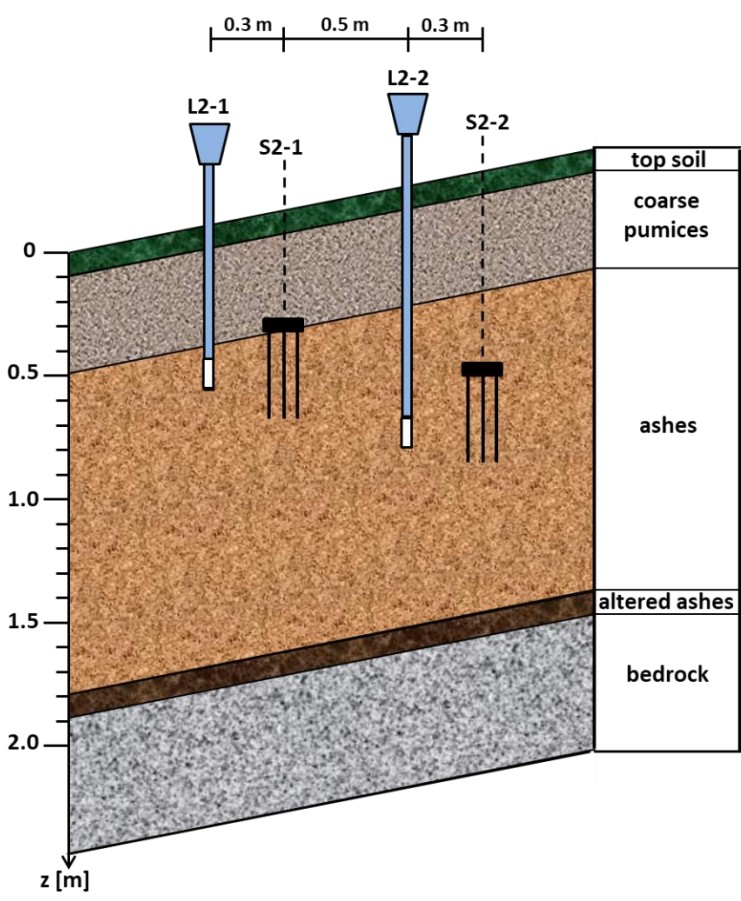

**Figure 5: Local stratigraphy of the monitored deposit and position of the "Jet-fill" tensiometers (L2-1 and L2-2), and of the TDR sensors (S2-1, S2-2). The ceramic tips of tensiometers, L2-1 (z = 0.60 m) and L2-2 (z = 1.00), are respectively installed at the same depth, z, of the centres of the TDR probes, S2-1 and S2-2.**

Regarding the water retention properties, Figure 6 shows the results of nine laboratory wetting tests performed by Damiano and Olivares (2010). These tests were carried out in the laboratory on a 40 cm thick model slope formed with volcanic ashes

taken from Cervinara and the nearby Monteforte Irpino sloping site. The model slope, reconstituted at the maximum field porosity of 75% (Table 1), was subjected to artificial precipitation. The $s$ and $\theta$ measurements were respectively provided by a miniaturised tensiometer and a TDR probe installed close to each other at depths from 1.5 to 8.5 cm. The initial $s$ and $\theta$ values were respectively in the range 15-60 kPa and 0.21-0.34 m$^3$/m$^3$ (Fig. 6). Once infiltration started, the experimental points of each test first ran rather a flat path and then, for $s$ smaller than 4 kPa, a more inclined path until full saturation. All the points

located along the steeper paths were fitted with the van Genuchten equation (1980)

$$\vartheta = \vartheta_r + \frac{\vartheta_s - \vartheta_r}{[1+(\alpha s)^n]^m} \tag{1},$$

where $\vartheta_s$ is the saturated volumetric water content, $\vartheta_r$ is the residual volumetric water content, $\alpha$, $n$ and $m$ are fitting parameters. Assuming $\theta_s = 0.75$ m$^3$/m$^3$ (that corresponds to the soil porosity), $\theta_r = 0$ (a value which is consistent with the

coarse-grained nature of the soil), and $m = \frac{n-1}{n}$ (according to Mualem, 1976), Table 2 shows the best fitting $\alpha$ and $n$ values.

In order to help the interpretation of the in-situ hydrological response, the obtained curve will be assumed as a possible reference main wetting curve.

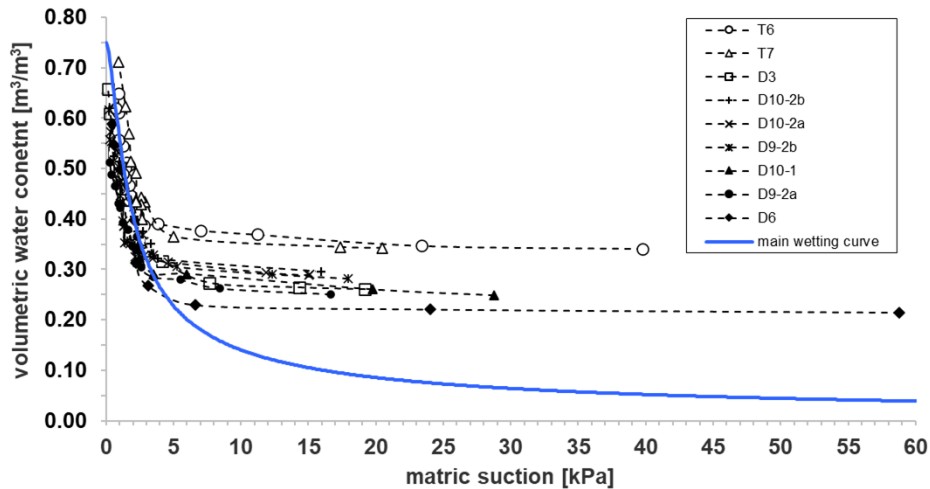

**Figure 6: Coupled values of matric suction and volumetric water content measured during infiltration tests (Damiano and Olivares, 2010). The tests acronyms shown in the legend are those reported in the repository dataset (Comegna et al., 2020). The main wetting**
**curve is obtained fitting all the experimental points featured by matric suction values lower than 4 kPa with the van Genuchten Eq. (1). The fitted van Genuchten parameters are shown by Table 2.**

**Table 2: Van Genuchten parameters (Eq. 1), fitted to the main wetting curve shown in Figure 6: saturated volumetric water content, $\theta_s$; residual volumetric water content, $\theta_r$; parameters $\alpha$, $n$ and $m$.**

| $\vartheta_s$ [m³/m³] | $\vartheta_r$ [m³/m³] | $\alpha$ [kPa⁻¹] | $n$ [-] | $m$ [-] |
|---|---|---|---|---|
| 0.75 | 0.00 | 1.00 | 1.72 | 0.42 |

**2.3 Monitoring station**

At the end of 2009, an automatic monitoring station was installed at the elevation of 560 m a.s.l., next to the right side of the landslide triggered in 1999 (Fig. 3b), to have continuous information about rainfall depth, matric suction and volumetric water
content (Comegna et al. 2016a). Figure 7a shows a schematic representation of the installed devices, whose main features are reported by Table 3. Rainfall was automatically recorded at a hourly time step by a rain gauge (Fig. 7b) having a sensitivity of 0.2 mm. Suction was measured by two "Jet-fill" tensiometers, equipped with tension transducers (Fig. 7c). The 7 cm long ceramic tips of the tensiometers, named L2-1 and L2-2 (Fig. 5), were pushed into the soil, at the depths z = 0.60 m and z = 1.00 m, through small holes previously dug by a drill. The uppermost part of the hole was then filled with a bentonite–cement
mixture to avoid any water infiltration. A careful maintenance was granted by regularly checking the complete saturation of the instruments (especially after long-lasting dry periods) and filling the tube with de-aired water in order to remove air

bubbles. Moreover, the instruments were carefully checked during the coldest periods, featured by temperature lower than 0°C, when the de-aired water contained in the upper part of the hydraulic circuit of the tensiometers could freeze, thus affecting the correctness of the pressure transducer reading. If it occurred, we made note of that in order to give a correct interpretation of the corresponding registered data. During the cold periods we also controlled that the expanded volume of ice did not break either the pressure transducer or the plexiglass tensiometer tubes. Volumetric water content was measured by two probes for Time Domain Reflectometry (TDR), named S2-1 and S2-2 (Fig. 5), that were installed in a row with the tensiometers, at a distance of 30 cm from them. Each sensor consists of three 40 cm long metallic rods (Fig. 7d), having a diameter of 3 mm and spacing of 15 mm.  The centres of S2-1 and S2-2 were respectively installed at the same depth of the ceramic tips of L2-1 and L2-2 (Fig. 5). Once vertically buried in the soil, the TDR probes were connected through coaxial cables and a multiplexer to a Campbell Scientific Inc. TDR-100 reflectometer (Fig. 7e). TDR readings provide the soil bulk dielectric permittivity, $\varepsilon_r$, which can be converted to $\theta$ through a calibration relationship (Topp et al., 1980). A specific relationship was purposely found by Guida et al. (2012) through targeted laboratory tests on undisturbed samples taken nearby the monitoring station; the average error in the estimation of $\theta$ is ±0.02 m³/m³. All the sensors were connected to the Campbell Data Logger located about 5 m far (Fig. 7e). The monitoring station was powered by a solar panel with a 12V backup battery. The automatic acquisition and storage of data was set with a time resolution of six hours.

Hourly air temperature data are also available, being provided by the Pietrastornina weather station (located at 495 m a.s.l. and 15 km from Cervinara), that is managed by the "Functional Centre for forecast, prevention and monitoring of risks and alerting for civil protection" of Campania Civil Protection Agency.

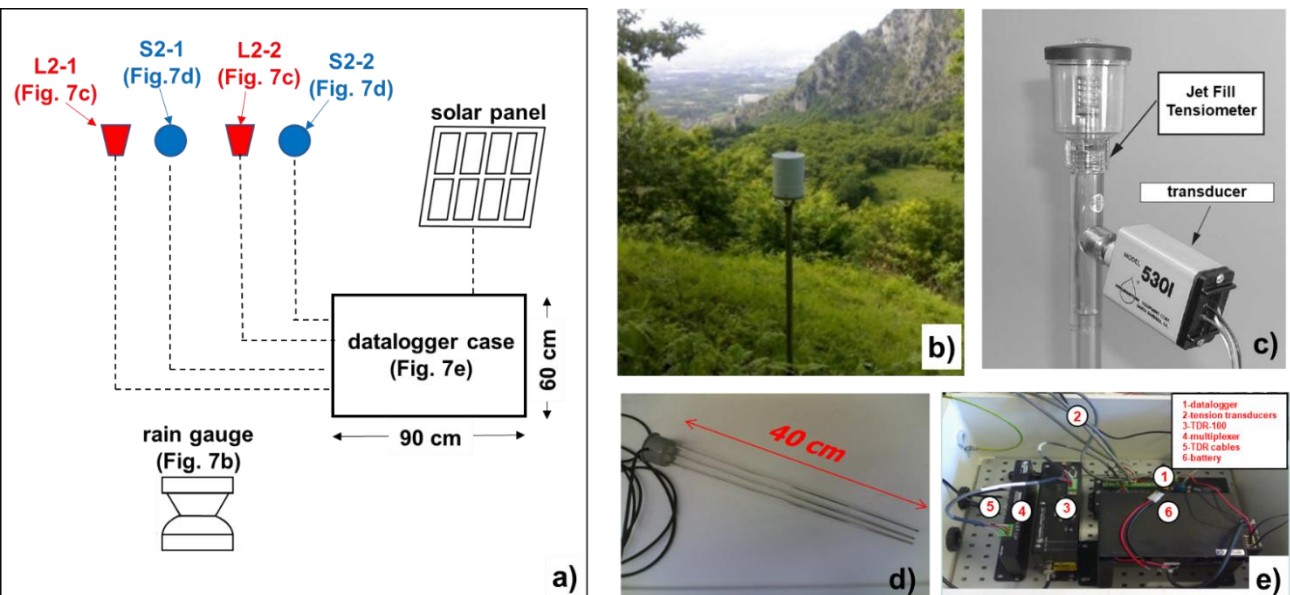

**Figure 7: Installed instruments: a) schematic representation; b) rain gauge; c) "Jet-fill" tensiometer; d) TDR probe;  d) datalogger case. The main features of the electronic instruments are reported by Table 3.**

**Table 3: Main characteristics of the installed electronic instruments.**

| ELETRONIC SENSORS | COMPANY | MODEL | MEASUREMENT RANGE | TEMPERATURE RANGE | ACCURACY |
|---|---|---|---|---|---|
| Data logger and data Acquisition System | Campbell Scientific Inc. (Logan, UT, USA) | CR-1000 | - | -25°C to +50°C | ±(0.06% of reading +offset) at 0 °C to 40 °C for Analog Voltage |
| Time Domain Reflectometer | Campbell Scientific Inc. (Logan, UT, USA) | TDR-100 | -2 m to 2100 m (distance) and 0 to 7µs (time) | -40°C to +55°C | ±0.02 m³/m³ |
| Multiplexer for TDR System | Campbell Scientific Inc. (Logan, UT, USA) | SDM8 x 50 | 8-Channel | -40°C to +55°C | not defined |
| Tension Transducer | Soil Moisture Equipment Corp. (Goleta, CA, USA) | 5301 | 0–85 cBar | 0°C to +60°C | 0.25% |
| Rain gauge | Campbell Scientific Inc. (Logan, UT, USA) | ARG100 | 0–500 mm/hr | - | 98% @ 20 mm/hr, 96% @ 50 mm/hr, 95% @ 120mm/hr |


The monitored $s$ and $\theta$ data have been then used to carry out some analyses aimed to estimate the slope stability conditions. In particular, the factor of safety FS (z; t) at depth $z$ and time $t$ has been calculated by the following formula provided by the simplified infinite slope model, that is suitable for Cervinara slope (Greco et al., 2013; Comegna et al., 2016b),

$$FS(z;t) = \frac{\tan\varphi'}{\tan\alpha} + \frac{c_{app}(z;t)}{\gamma \cdot z \cdot sen\alpha \cdot cos\alpha} \tag{2},$$

where $\varphi'$ is the friction angle, $\alpha$ is the slope angle, $\gamma$ is the unit weight of the deposit (assumed homogeneous), while $c_{app}$, known as *apparent cohesion*, is a strength component changing with time according to the $s$ and $\theta$ variations. Vanapalli et al. (1996) provide the following expression for $c_{app}$

$$c_{app}(z;t) = s(z;t) \cdot \frac{\vartheta(z;t)-\vartheta_r}{\vartheta_s-\vartheta_r} \cdot \tan\varphi' \tag{3}.$$

Therefore, $c_{app}$, and consequently FS, decreases with $s$ due to infiltration. Moreover, being an assigned FS obtained by different couples of $s$ and $\theta$, it's possible to plot on a water retention plane different iso-FS curves, that could help to determine the
current slope stability conditions.

The following Section 3 describes the field data collected from January, 1[st], 2011 to January, 31[st], 2012: this period has been chosen because of the abundance of data useful for a correct interpretation of the annual hydrological response.

Section 4 reports some considerations concerning the influence of the monitored hydraulic hysteresis on the slope stability conditions.

## 3 Results of field monitoring

Figure 8 shows the monthly cumulative precipitation in 2011 provided by the rain gauge installed on the slope. The total cumulative rainfall was 1360 mm, a value lower than the mean yearly rainfall in the same area (Figure 4). A daily precipitation higher than 1 mm was recorded 99 times; the daily rainfall exceeded the value of 50 mm only in five cases (February, 16th; April, 30th; September, 19th; November, 6th; December, 5th). March was the rainiest month, with a total precipitation of 296 mm, i.e. 22% of the yearly rainfall. The dry season started in June continuing until the end of October. In this season some significant isolated rainy events characterised by daily precipitation ranging between 17.4 mm and 19.2 mm occurred on July, 29th; September, 7th; October, 8th; October, 22nd; another more severe event totaling 53 mm took place on September, 19th. In the time interval November – December, the cumulative rainfall was about 30% of the annual precipitation: the most intense daily rainfall occurred on November, 6th (58 mm). Figure 8 also shows the average monthly values of minimum, daily mean, and maximum temperature monitored at a hourly scale by the Pietrastornina weather station. The daily mean air temperature was close to average, with a slightly warmer summer. The potential evapotranspiration can hence be assumed to be close to the average estimated values shown in Figure 4. In particular, the mean daily temperature was higher than 15°C from May, 19th, to October, 7th, attaining values higher than 20°C from July, 30th, to September, 18th. The lowest and highest average values were respectively measured in February (4 °C), and in August (31 °C). Although some discrepancies due to the distance of the Pietrastornina weather station from the investigated slope, the temperature date above elaborated can be considered reliable.

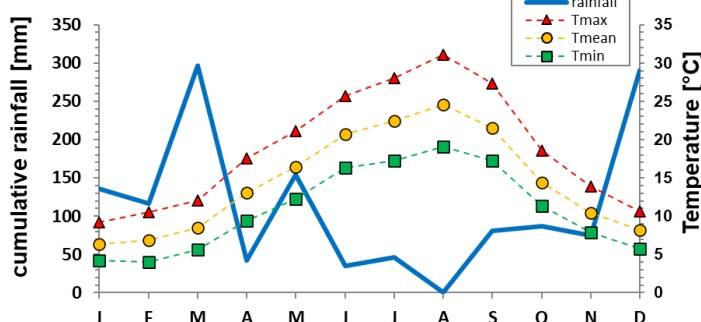

**Figure 8: Monthly cumulative rainfall and average monthly minimum ($T_{min}$), daily mean ($T_{mean}$) and maximum ($T_{max}$) values of air temperature monitored in 2011. $T_{min}$, $T_{mean}$ and $T_{max}$ are calculated with the data monitored by the Pietrastornina weather station managed by the "Functional Centre for forecast, prevention and monitoring of risks and alerting for civil protection" of Campania Civil Protection Agency.**

The information obtained by coupling $\theta$ and $s$ data monitored at both the investigated depths of 0.60 m and 1.00 m are discussed in the following subsections by distinguishing eight time windows characterised by different weather conditions. For the sake of clarity, it has to be pointed out the tensiometers and the TDR probes sample different soil volumes; this might lead to an imperfect matching of data. Due to some technical problems, related to the emptying of the tensiometers (occurring especially during the warmest periods) or to some loss in battery power, the records present some missing data. In particular, unfortunately no retention data are available from July, 19th to November, 4th, 2011 and from December, 6th, 2011 to January,

$6^{th}$, 2012. Despite the gap, it allowed to recognize important aspects of the hysteretic hydrologic response of the investigated deposit.

### 3.1 Time window A-B: January, $1^{st}$ - May, $8^{th}$ 2011

The total precipitation in this time interval was 695 mm, which corresponds to 51% of the annual cumulative value. Until March, $12^{th}$, the mean daily air temperature ranges between 0.5°C and 12.8°C (Fig. 9a), then it steadily increases from 5.8°C to 16.7°C (with an increasing trend of 1.8°C/month).

On January, $1^{st}$, the $\theta$ and $s$ values measured by S2-1 and L2-1 at depth of 0.60 m are respectively 0.39 $m^3/m^3$ and 11 kPa (Fig. 9b). These values are the result of the antecedent weather conditions; in particular, the total precipitation in the previous 30 days had been 170 mm and the mean daily air temperature 7°C (Fig. 9a). As shown in Figure 9b, the $\theta$ measured in the examined window is 0.34-0.41 $m^3/m^3$ while $s$ ranges in the interval 3-16 kPa. During the dry days, $\theta$ decreases with a rate of -1%/month while $s$ tends to increase with an average rate of about 1.4 kPa/month. This drying path is periodically reversed by some rainfall-induced wetting processes. In particular, three sudden $s$ drops are recorded on January, $23^{rd}$, February, $17^{th}$, and May, $1^{st}$, due to very similar rainfall events featured by a total precipitation of 48-58 mm cumulated over the antecedent 48 hours.

At depth of 1.00 m, $\theta$, measured by S2-2, and $s$, measured by L2-2, range respectively in the intervals 0.29-0.37 $m^3/m^3$ and 4-16 kPa (Fig. 9c). $\theta$ tends to decrease by -0.8%/month, while $s$ is increasing with an average rate of about 1.0 kPa/month. These trends are hence slower than at the shallower depth. This reflects a minor role of evapotranspiration during winter and early spring (when the vegetation is leafless). In this period, the soil profile is mainly drained downward by gravity, except during rainfall, when capillary gradient favours rapid infiltration. The small variations of $\theta$ and $s$ at both depths, despite the large amount of precipitation, suggest that the soil cover is being crossed by an intense downward flux without being retained (or being retained only in a little part). Rainfall events can cause just temporary small increments of $\theta$ followed by slower reductions. Indeed, at both depths $\theta$ is steadily higher than the field capacity (i.e. about 0.25 $m^3/m^3$).

All data have been reported in the water retention plane $s$-$\theta$, shown in Figure 10a and 10b, together with the corresponding manual fitting curves, named AB. At both investigated depths, the curve AB is quite flat with an overall slope of about -0.4 %/kPa.

### 3.2 Time window B-C: May, 8th - June, 22nd 2011

This window is featured by a cumulative rainfall of 85 mm and a daily air temperature ranging in the interval 10-24°C, with an increasing trend of 4.3°C/month (Fig. 9a). During this season, vegetation starts flourishing thus accommodating the increasing evapotranspiration demand and influencing the hydrological soil response through root water uptake. The actual evapotranspiration may approach the limiting PET value, but very likely never reaching it because the atmospheric demand is not fulfilled by soil water.

At depth z = 0.60 m, $s$ ranges between 10 kPa and 24 kPa, growing with a rate of about 9 kPa/month. This matches a $\theta$ reduction, that reaches the value of 0.25 m$^3$/m$^3$ with a decreasing trend of -9 %/month (Fig. 9b), which is more pronounced than during the previous time window due to an intense root water uptake from the uppermost soil layer. Collected data were fitted by the curve BC in Figure 10a provided by Eq. (1) identifying the best fitting $\alpha$ and $n$ parameters shown in Table 4 (assuming again $\theta_s$ = 0.75 m$^3$/m$^3$, $\theta_r$ = 0, and $m = \frac{n-1}{n}$). The path BC is clearly steeper than the curve AB.

Table 4: Van Genuchten parameters (Eq. 1) fitted to the curve BCD shown by Figure 10a: saturated volumetric water content, $\theta_s$; residual volumetric water content, $\theta_r$; parameters, $\alpha$, $n$ and $m$.

| $\vartheta_s$ [m$^3$/m$^3$] | $\vartheta_r$ [m$^3$/m$^3$] | $\alpha$ [kPa$^{-1}$] | $n$ [-] | $m$ [-] |
|---|---|---|---|---|
| 0.75 | 0.00 | 0.11 | 2.17 | 0.54 |

At depth z = 1.00 m (Fig. 9c), until June, 6$^{th}$, $\theta$ and $s$ display little variations, moving respectively from 0.33 m$^3$/m$^3$ to 0.31 m$^3$/m$^3$ and from 15 kPa to 17 kPa. Measured values are again well fitted by the curve AB (Fig. 10b). After this period, which is probably still characterised by some gravitational downward drainage, the soil starts drying quickly at this depth too being forced by root water uptake. The water content decreases with a rate of about -4%/month attaining a value of 0.27 m$^3$/m$^3$ at the end of this time window, while the increasing $s$ rate is similar to the one observed at 0.60 m. These data are well fitted by the path CD (Fig. 10b), which is steeper than the path AB, but gentler than the curve BCD detected at 0.60 m because of a lower evapotranspiration effect. The fitting parameters are reported in Table 5. Besides to evapotranspiration effects, the different responses observed at the two depths might be justified also by small differences in grain size and/or void ratio of the soil (Comegna et al., 2016a).

Table 5: Van Genuchten parameters (Eq. 1) fitted to the curve CD shown by Figure 10b: saturated volumetric water content, $\theta_s$; residual volumetric water content, $\theta_r$; parameters, $\alpha$, $n$ and $m$.

| $\vartheta_s$ [m$^3$/m$^3$] | $\vartheta_r$ [m$^3$/m$^3$] | $\alpha$ [kPa$^{-1}$] | $n$ [-] | $m$ [-] |
|---|---|---|---|---|
| 0.75 | 0.00 | 0.76 | 1.33 | 0.25 |

### 3.3 Time window C-D: June, 22$^{nd}$ - July, 18$^{th}$ 2011

In this dry time interval the average daily temperature is 23.7°C with an increasing trend of 3.3°C/month and the cumulative rainfall is 14 mm (Fig. 9a). The flourishing vegetation and the high temperature suggest that evapotranspiration largely exceeds infiltration by rainwater, causing drainage of the soil cover.

At z = 0.60 m, $\theta$ reaches a value of 0.17 m$^3$/m$^3$, while $s$ grows by about 22 kPa/month until a value of 35 kPa (Fig. 9b). It is worth noting that in the retention plane the path BC can properly fit recorded field data (Fig. 10a). At z = 1.0 m, $\theta$ reaches the value of 0.24 m$^3$/m$^3$, while $s$ increases with a rate of about 13 kPa/month attaining a value of 34 kPa (Fig. 9c). In the retention plane, the field data are well interpolated by the curve CD (Fig. 10b).

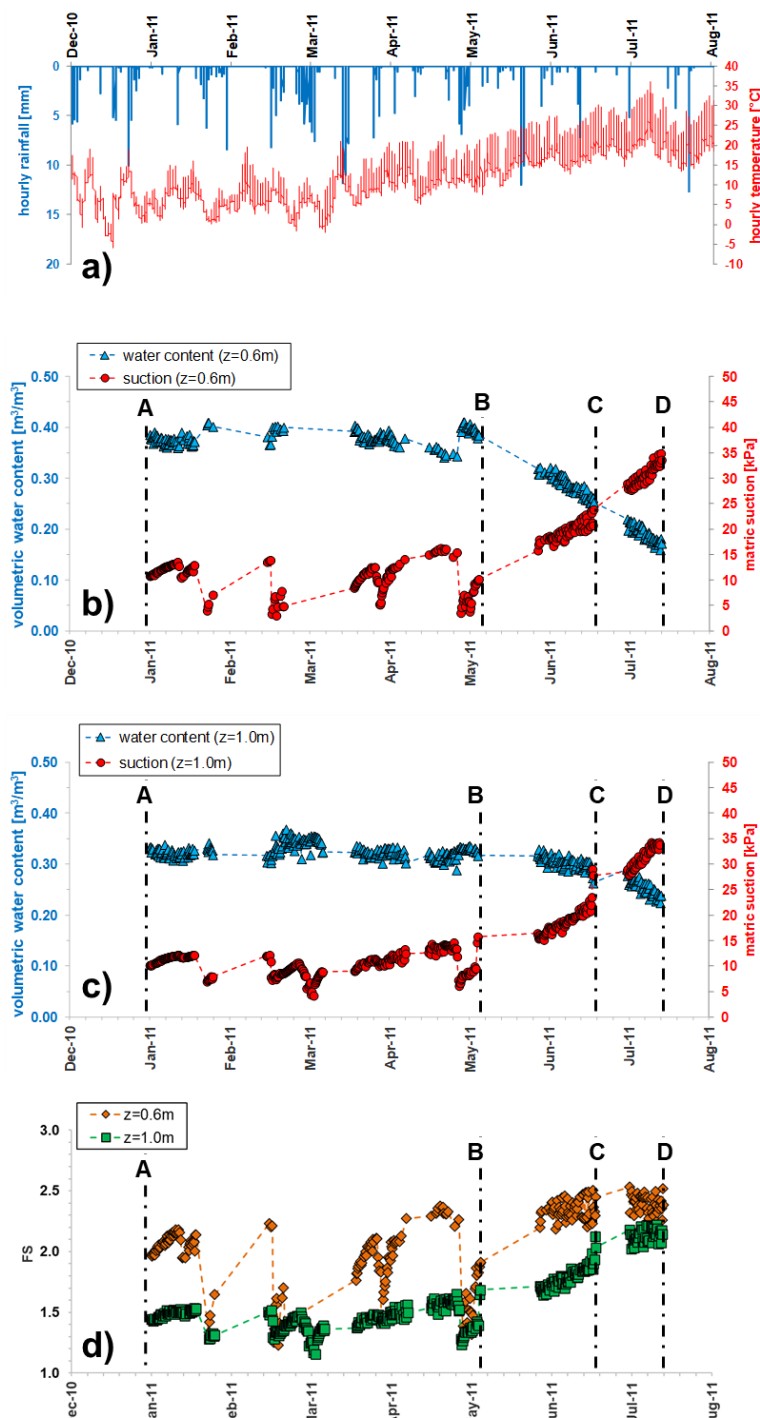

Figure 9: Hourly weather data (a), matric suction and volumetric water content monitored from January to July, 2011 at depth z = 0.60 m (b) and z = 1.00 (c), factor of safety, FS (d), calculated by Eq. (2), provided by the simplified infinite slope model, assuming a homogeneous deposit with slope angle α = 40°, unit weight γ = 14 kN/m³, cohesion c' = 0, friction angle $\varphi'$ = 38°, $\theta_s$ = 0.75 m³/m³, $\theta_r$ = 0.

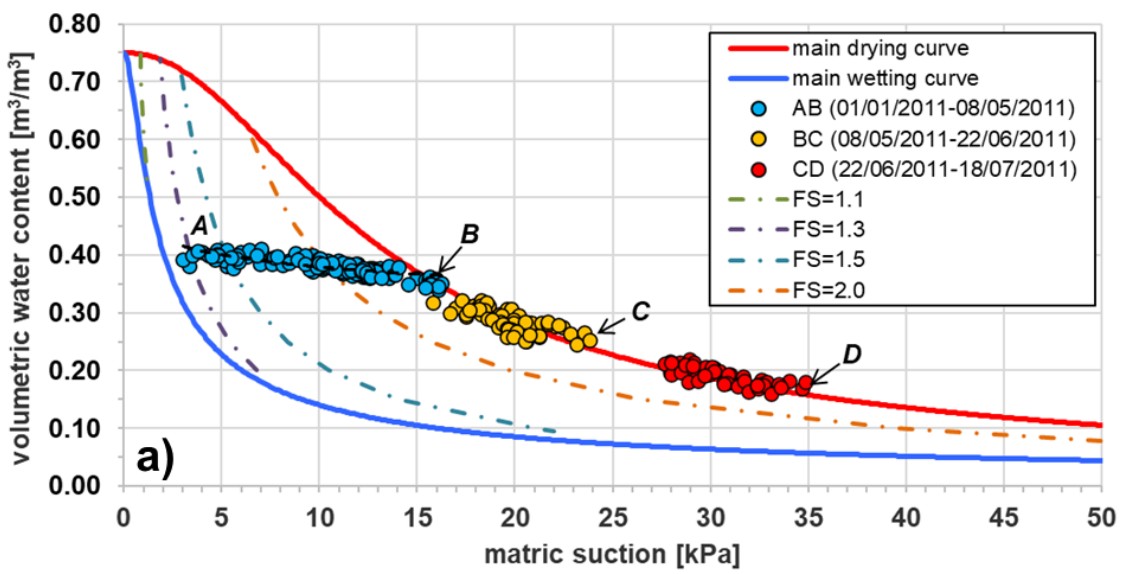

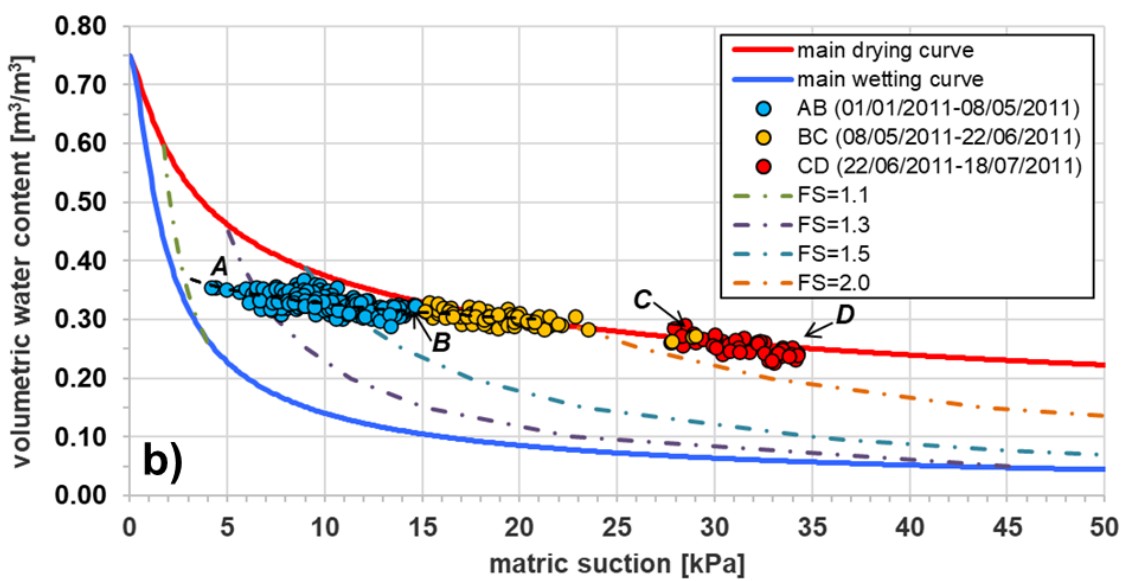

**Figure 10: Volumetric water content and matric suction recorded from January to July, 2011 at depths z = 0.60 m (a) and z = 1.00 m (b) and iso-Safety Factor curves. The main drying curve at z = 0.60 m is obtained fitting with the Eq. (1) all the experimental points monitored from 08/05/2011 to 18/07/2011 (the fitted parameters are shown by Table 4). The main drying curve at z = 1.00 m is obtained fitting with the Eq. (1) all the experimental points monitored from 07/06/2011 to 18/07/2011 (the fitted parameters are shown by Table 5). The main wetting curves at both depths coincide with that derived by the infiltration tests (Damiano and Olivares, 2010) shown in Figure 6. The safety factor, FS, is calculated by Eq. (2), provided by the simplified infinite slope model, assuming a homogeneous deposit with slope angle α = 40°, unit weight γ = 14 kN/m³, cohesion c' = 0, friction angle φ' = 38°, $\theta_s$ = 0.75 m³/m³, $\theta_r$ = 0.**

### 3.4 Time window E-F: November, 5th - November, 6th 2011

In the time interval from July, 19th to November, 4th, during which, as already stated, monitoring of $\theta$ and $s$ stops, the air temperature goes from the mean value of 24.6°C, reached in August, to 14.4°C, in October. Regarding precipitation, from September to October the rain gauge records a cumulative precipitation of 168 mm fallen in only 9 isolated rainy days (Fig. 11a). Such a few concentrated precipitation seem to have been recorded by the shallowest sensors only. In fact, if the data acquired on November, 5th, are compared to those monitored on July, 18th, we can notice a $\theta$ increase (from 0.17 to 0.24 m³/m³)

and a $s$ decrease (from 35 to 22 kPa) at 0.60 m (Fig. 11b), while the sensors at z = 1.00 m (Fig. 11c) record a small water content decrease (from 0.24 to 0.23 m³/m³) and a $s$ increase, from 34 to 47 kPa.

The most intense daily rainfall in 2011 takes place on November, 6th. The total precipitation is 58 mm (Fig. 11a) and causes a $\theta$ increase from 0.24 to 0.43 m³/m³ (Fig. 11b) at 0.6 m, and from 0.23 to 0.27 m³/m³ at 1.0 m (Fig. 11c). At both depths the highest drop in $s$ is recorded; in fact, the decrease measured by L2-1 is from 22 kPa to 1.9 kPa (Fig. 11b) and the one recorded

by L2-2 is from 47 kPa to 13 kPa (Fig. 11c). These data are represented in the water retention plane by the paths EF (Figs. 12a and 12b) well different from those ran in May and June. It's worth noting that at the shallowest depth, the final point F reaches the main wetting curve obtained by interpolating the flume tests described in Section 2 (Fig. 5).

### 3.5 Time window F-G: November, 6th – December, 3rd 2011

This time interval is characterised by dry weather. In fact, one single rainfall event only of 12 mm occurs on November, 22nd. The mean temperature is about 10°C. During this period, when the leaves of deciduous trees fall, the vegetation enters a dormant phase, during which they need very little water. Hence, the atmospheric evapotranspiration demand, that is small as typical of winter months, is likely larger than the amount of water actually extracted from the soil by the vegetation. An essentially downward flow, initially driven by a high potential gradient due to a wetter uppermost soil profile, is consequently

favoured then progressively approaching a slow gravity driven drainage. In fact, at z = 0.60 m, $\theta$ decreases from 0.43 to 0.28 m³/m³ while $s$ increases from 1.9 kPa to 20 kPa (Fig. 11b). In the water retention plane, the corresponding drying path FG is located above previous wetting path EF, about parallel to it (Fig. 12a). It is worth to note that at point G it reaches the BCD curve travelled from May to July, confirming that drying develops according to smoother paths and gently approaches the field capacity (about 0.25 m³/m³), when soil drainage is not forced by root water uptake.

At z = 1.00 m, $\theta$ decreases from 0.27 to 0.25 m³/m³ and $s$ increases from 13 kPa to 19 kPa (Fig. 11c). The corresponding drying path FG pursues backwards previous path EF (Fig. 12b).

### 3.6 Time window G-H: December, 3rd - December, 5th 2011

On December, 5th, after a precipitation of 98 mm in 48 hours, $\theta$ increases at both depths. In particular, at z = 0.60 m $\theta$ grows from 0.28 to 0.40 m³/m³ and $s$ drops to 2.5 kPa (Fig. 11b). The wetting curve GH overlaps previous FG drying path (Fig. 12a).

Again, the final point H reaches the assumed main wetting curve.

At z = 1.00 m $\theta$ increases from 0.25 to 0.36 m$^3$/m$^3$, less than above, while $s$ decreases to 4.5 kPa. The final point H does not reach the assumed main wetting curve (Fig. 12b).

### 3.7 Time window H-I: December, 5$^{th}$ – December, 11$^{th}$ 2011

No precipitation occur during this short time window. Available data concern only depth z = 1.00 m. $\theta$ decreases from 0.36 to 350   0.31 m$^3$/m$^3$ and $s$ increases to 10 kPa (Fig. 11c). The drying path HI is located above and parallel to the wetting path GH (Fig. 12b).

### 3.8 Time window L-M: January, 7$^{th}$, 2011 – January, 31$^{st}$, 2012

This period is characterised by negligible evapotranspiration owing to cold temperatures and leafless vegetation. Hence, the observed $\theta$ and $s$ trends may be ascribed to gravitational downward drainage, which in the long run would lead the soil cover 355   to approach field capacity. Until January, 21$^{st}$, 2012 $s$ increases from 6 kPa to 13 kPa at z = 0.60 m and from 10 kPa to 14 kPa at z = 1.00 m. On that date, a 12-hours cumulative 16 mm rainfall causes a drop in $s$ of 6 kPa and 3 kPa respectively at the shallowest and deepest tensiometers. Then $s$ increases again until the final values of 12 kPa (z = 0.60 m) and 13 kPa (z = 1.00 m) associated with $\theta$ of 0.32 and 0.31 m$^3$/m$^3$. All field data are quite well interpolated by the paths GH at z = 0.60 m (Fig. 12a) and HI at z = 1.00 m (Fig. 12b).


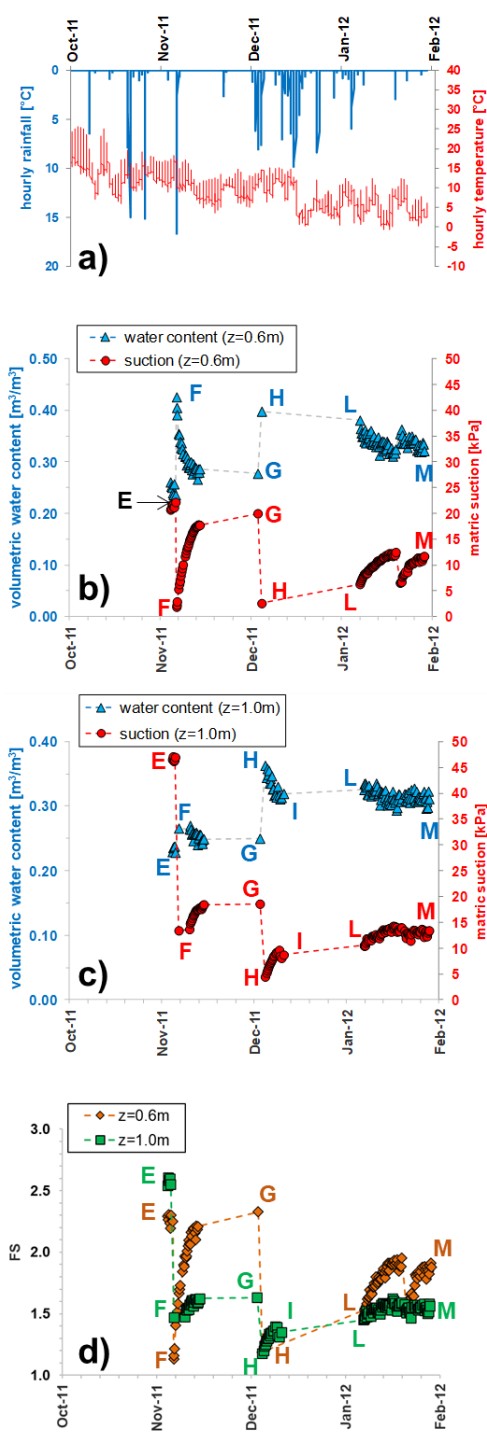

**Figure 11: Monitored hourly weather data (a), matric suction and volumetric water content from November, 2011 to January, 2012 at the depth z = 0.60 m (b) and z = 1.00 (c), factor of safety, FS (d), calculated by Eq. (2), provided by the simplified infinite slope model, assuming a homogeneous deposit with slope angle $\alpha$ = 40°, unit weight $\gamma$ = 14 kN/m³, cohesion c' = 0, friction angle $\varphi'$ = 38°, $\theta_s$ = 0.75 m³/m³, $\theta_r$ = 0.**

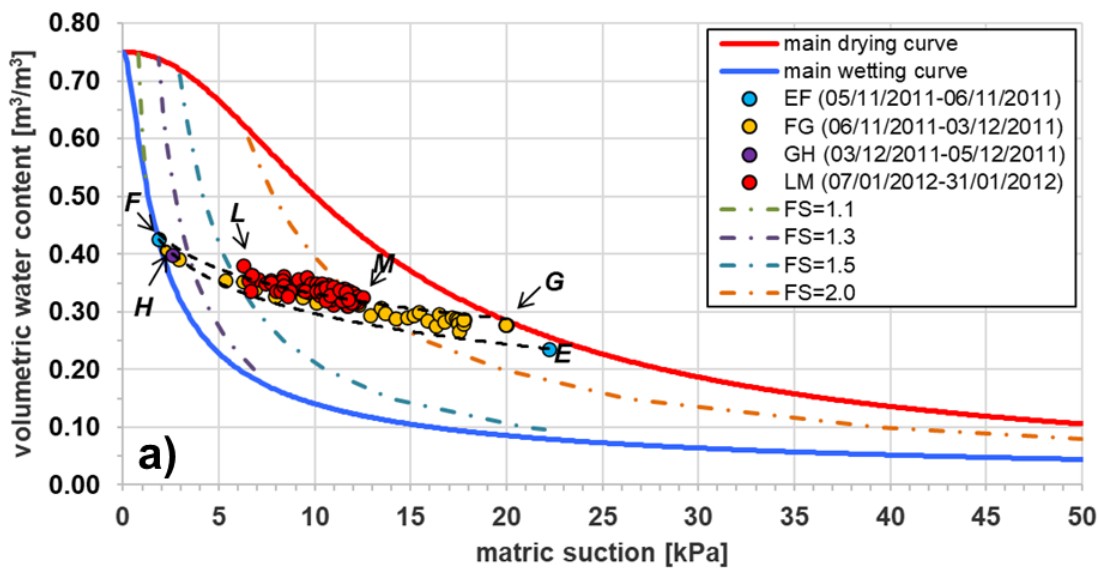

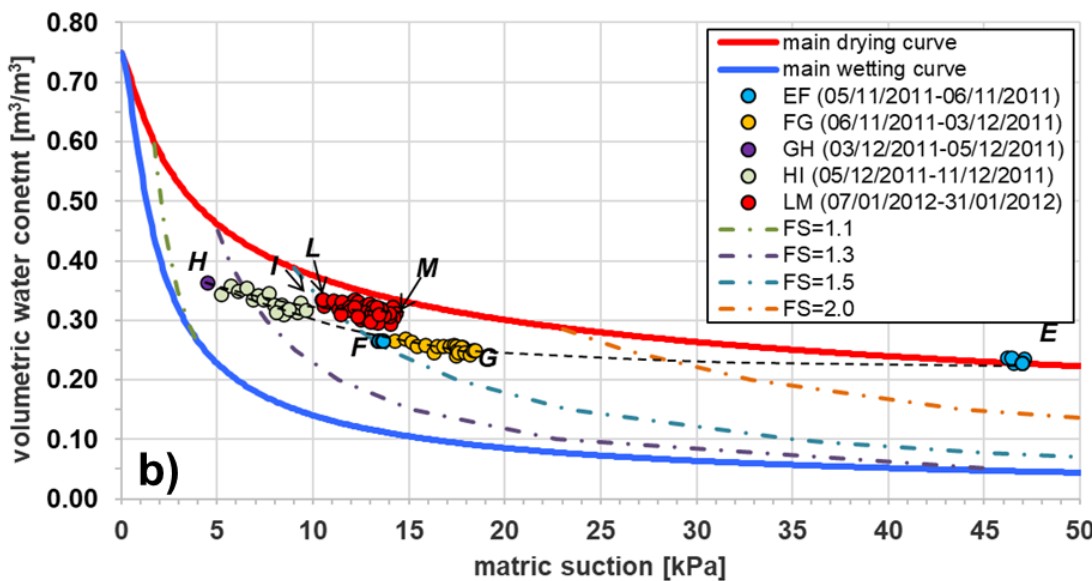

**Figure 12: Volumetric water content and matric suction, monitored from November, 2011 to January, 2012 at depths z = 0.60 m (a) and z = 1.00 m (b), and iso-Safety Factor curves. The main drying curve at z = 0.60 m is obtained fitting with the Eq. (1) all the experimental points monitored from 08/05/2011 to 18/07/2011 (the fitted parameters are shown by Table 4). The main drying curve at z = 1.00 m is obtained fitting with the Eq. (1) all the experimental points monitored from 07/06/2011 to 18/07/2011 (the fitted parameters are shown by Table 5). The main wetting curves at both depths coincide with that derived by the infiltration tests (Damiano and Olivares, 2010) shown in Figure 6. The safety factor, FS, has been calculated by Eq. (2), provided by the simplified infinite slope model, assuming a homogeneous deposit with slope angle $\alpha = 40°$, unit weight $\gamma = 14$ kN/m³, cohesion c' = 0, friction angle $\varphi' = 38°$, $\theta_s = 0.75$ m³/m³, $\theta_r = 0$.**

**4 Discussion**

The paths plotted in Figs. 10 and 12 show that, at different times, different values of $s$ have been observed at both instrumented depths for the same $\theta$. The relationship between these two variables is then not univocal. In particular, the difference depends on the initial conditions (i.e. on the starting point). This reveals the hysteretic nature of the hydrological soil response. Therefore, all obtained paths should be considered as scanning curves located between the main drying and the main wetting curve. In more detail, the steepest drying paths obtained during the warmest days as a result of intense evapotranspiration

owing to flourishing vegetation (curve BCD at z = 0.60 m; curve CD at z = 1.00 m) tend to the assumed main drying curve. It's also interesting to notice that, at 1.0 m depth, this final steeper path is reached on June, 6[th], i.e. with a delay of about one month with respect to the shallowest depth (where it has been attained on May, 8[th]). Such a result could be related to the delayed and mitigated effect of evapotranspiration due to the water uptake by roots, which are denser at depths less than 0.50 m, but are present down to a depth of 1.50 m more or less. During the periods of leafless vegetation and low temperatures,

when the amount of evapotranspiration is modest, the drying paths are less steep and well below the assumed upper boundary. According to our field data, the internal hysteresis is not particularly relevant. However, some little differences between the drying and the wetting scanning paths have been recognized at both depths when the highest $\theta$ is attained next to the main wetting curve. In particular, Figure 10a shows that the scanning path observed at depth z = 0.60 m after the rainfall event finished on November, 6[th] (curve FG) is above that previously monitored (curve EF); Figure 10b shows that the scanning path

observed at depth z = 1.00 m after the rainfall event completed on December, 5[th] (curve HM) is above the one monitored before that (curve EH).

For the sake of clarity, it has to be pointed out the tensiometers and the TDR probes sample different soil volumes; this might lead to an imperfect matching of data, i.e. any variation in $s$ and in $\theta$ is not simultaneously detected by the sensors. For instance, if the wetting front advances downward, the 40 cm long TDR probe can detect it with some advance compared to the ceramic

tip of the tensiometers (the centres of the two sensors are aligned, so the upper edge of the TDR probe is above the upper edge of the tip). The temporal mismatch will be larger for steeper wetting front. A similar issue would affect the edges of the sensors during vertical (gravity-driven) drainage processes, while soil drying caused by root uptake is expected to be more evenly distributed throughout the entire root zone. These issues would certainly affect the coupling especially in the initial stage of infiltration and drainage process, which are characterized by steeper gradients. However, the measurements are acquired every

six hours, and looking at Figures 9 and 11, it appears that $s$ and $\theta$ variations at the two investigated depths, 40 cm apart from each other, are detected nearly simultaneously. Hence, it is expected that the temporal mismatch may affect only one or two measurement points during each of the wetting/drying paths discussed in Figures 10 and 11, which consist of many measurement points as they refer to long lasting processes.

In order to estimate to what extent the slope stability conditions are affected by the hysteretic response, some simple analyses

have been carried out by the infinite slope model. In particular, the factor of safety FS at depth z has been calculated by Eq. (2). Assuming a homogeneous deposit with slope angle $\alpha$ = 40°, unit weight $\gamma$ = 14 kN/m$^3$, cohesion c' = 0, friction angle $\varphi'$

= 38°, $\theta_s$ = 0.75 m³/m³, $\theta_r$ = 0, the variation of FS with time is only due to fluctuations of the apparent cohesion, $c_{app}$. In particular, being φ' lower than α, FS remains higher than one only if $c_{app}$ is higher than a threshold value that could be calculated by Eq. (2). Regarding the examined case, at depths z = 0.60 m and z = 1.00 m, slope stability is guaranteed respectively by $c_{app}$ > 0.3 kPa and $c_{app}$ > 0.5 kPa.

Figures 9d and 11d show the fluctuations of FS during the period of monitoring. At z = 0.60 m, FS ranges between a minimum value of 1.13, attained on November, 6[th] (Fig. 11d), and a maximum of 2.22, on July, 5[th] (Fig. 9d), which respectively correspond to $c_{app}$ values of 0.8 and 6.6 kPa. At z = 1.00 m, FS ranges between 1.18 and 2.60, on December, 5[th], and on November, 5[th] (Fig. 11d), corresponding to a $c_{app}$ interval 1.5-11.6 kPa. The higher fluctuation of FS at the shallowest depth, z = 0.60 m is obviously due to a higher $s$ variation.

In Figures 10 and 12 the iso-Safety Factor curves, i.e. featured by constant FS values, have been plotted. For a given $s$, FS increases with $\theta$: this means that the lower scanning curves correspond to worse safety conditions. For instance, looking at Figure 12, the FS values calculated along the wetting path EF, that originates after the dry season, are lower than those corresponding to the drying curve LM that starts after the rainfall events occurred in November and in December; also, the changing rate of FS is always remarkable along the scanning paths where little $\theta$ changes can induce high $s$ changes.

It is interesting to notice that the lowest FS value is attained at z = 0.60 m on November, 6[th], i.e. after the most intense rainfall event recorded in 2011, when the wetting path reaches the assumed lower boundary at point F (Fig. 12a), featured by $s$ = 1.9 kPa and $\theta$ = 0.43 m³/m³. Starting from this condition, a further, just hypothetical, persistent and intense rainfall event could have forced the point to follow the final steeper branch of the main wetting curve. In particular, the failure condition (FS = 1) would have been reached for $\theta$ = 0.69 m³/m³ ($s$ = 0.40 kPa), i.e. for an increase $\Delta\theta$ = 0.26 m³/m³ (or a decrease $\Delta s$ = -1.5 kPa). Such a large increase in the water content indicates that landsliding in the area at hand is not so obvious, being the consequence of exceptional weather conditions as chronicles and statistical analyses suggest (Comegna et al., 2017).

## 5 Conclusions

The setup of an automatic field station allowed monitoring the annual cyclic hydrological response of a sloping deposit in pyroclastic air-fall soils. Even though the relationship between measured volumetric water content and suction values has to be carefully considered accounting for all the factors which can adversely affect its validity (small differences in grain size or porosity, which is dependent also on the installation procedures of the sensors into the soil; different soil volumes affecting the response of sensors; different reliability in data interpretation), monitoring provided useful information about the hydrological soil response.

In particular, collected data highlight the influence of the initial conditions, which depend on the antecedent wetting/drying history, on the weather-induced hydraulic paths. In fact, different volumetric water contents can be associated with the same matric suction due to the hysteretic soil response. Moreover, soil drying may be affected by evapotranspiration due to water extraction by roots, which varies throughout the seasons.

As indicated by simple stability analyses, in the examined period the slope has been far from failure conditions. In particular, the hydraulic path leading to slope failure should have featured by quite high volumetric water content changes. This are well detectable by TDR sensors, but are characterised by so low suction changes (less than 2 kPa) to be hardly measurable by ordinary tensiometers. These results unavoidably raise some questions on the best way to set up reliable early warning systems in areas threatened by rapid landslides in shallow unsaturated granular soil covers. These systems should be indeed based on: i) a monitoring system able to provide real time updates about the weather-induced hydraulic paths, and ii) a forecasting model accounting for the soil water retention properties. Both should be supported by the knowledge of the main drying and wetting curves that bound the water retention domain.

**Data availability**

The datasets, freely downloadable from https://doi.org/10.5281/zenodo.4281166 (Comegna et al., 2020), are provided through the following five separate Excel files:

- 1_Field data_ rainfall, containing the hourly rainfall measured in the considered time period (January, $1^{st}$, 2011 – January, $31^{st}$, 2012) by the rain gauge;
- 2_Field data_ temperature, containing the hourly temperature measured in the considered time period (January, $1^{st}$, 2011 – January, $31^{st}$, 2012) by the weather station located in the town of Pietrastornina and managed by the regional Civil Protection Agency;
- 3_Field data_suction & moisture content_z=0.6m, containing $s$ and $\theta$ measured in the considered time period (January, $1^{st}$, 2011 – January, $31^{st}$, 2012) at the depth z = 0.60 m respectively by tensiometer L2-1 and TDR probe S2-1 at a time resolution of 6 hours;
- 4_Field data_suction & moisture content_z=1.0m, containing $s$ and $\theta$ measured in the considered time period (January, $1^{st}$, 2011 – January, $31^{st}$, 2012) at the depth z = 0.60 m respectively by tensiometer L2-2 and TDR probe S2-2 at a time resolution of 6 hours;
- 5_Flume infiltration tests, containing the $s$ and $\theta$ measured during nine flume tests induced by an artificial rainwater infiltration, carried out by Damiano and Olivares (2010).

**Author contribution**

LC, ED, RG and LO installed the automatic monitoring station and took care about the maintenance of the instruments. LC analysed the monitored field data. ED and LO analysed the results of the laboratory infiltration tests to assume a reliable main wetting curve. LC and LP jointly conceived and set up the paper, discussing the issues with the other three authors. RG provided considerations about the role of vegetation. The contributions of the authors are equal.

**Competing interests**

The authors declare that they have no conflict of interest.

**Acknowledgements**

The research has been developed with the support of the project VALERE 2019, financed by the Università degli Studi della Campania "Luigi Vanvitelli".

The "Functional Centre for forecast, prevention and monitoring of risks and alerting for civil protection" of Campania Civil Protection Agency is gratefully acknowledged for providing temperature data.

The Authors thank the four anonymous Reviewers for their insightful comments and suggestions that improved the quality of the manuscript.

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
