# Peer review of "The hysteretic response of a shallow pyroclastic deposit"

_Earth System Science Data, 2020_

## Referee Comment (RC1)

**Manuscript:** *The hysteretic response of a shallow pyroclastic deposit.*

**Overview comments:**

This research paper presents a dataset of hydrological measurements carried out from 2011 to 2012 in a shallow deposit of loose pyroclastic soils located in Campania, Southern Italy.

The work is well written and illustrated with figures and tables, and their conclusions seem to be consistent with the results and discussion. However, the restructuration of some parts of the paper is considered appropriate (see 'line-by-line commentary'). Authors included data and methods in the 'Monitoring site' section. An independent and clear section for 'Data and methods' may be very appropriate, especially in a data description paper like this.

General vote: manuscript may become acceptable after major changes, which are detailed in this review.

**Line-by-line commentary:**

Line 10: I suggest to change 'had been involved' by 'was involved'.

Lines 18: (e.g., Mualem, 1976…).

Lines 22: 'specimen' instead of 'sample'?

Lines 30, 143, 150, 159, 207, 254, 278, 351, Figs. 8 and 10: Please, use the same style of English spelling along the manuscript (American or British English spelling). In some cases, authors have used British English spelling.

Line 52: Remove 'fully'?

Line 55: 'was' instead of 'had been'.

Line 77: (e.g., Fiorillo et al., 2001...).

Line 92: I propose to define a 'Data and methods' section in order to describe data and methods relative to the analysis of the monitored deposit from 2011 to 2012, whose results are presented in this work.

Line 119: Consider to put this paragraph at the end of the 'Data and methods' section.

Line 131: Please, explain what is each variable in the manuscript.

Line 194: Remove 'further and further'?

Line 342: Consider to re-write the sentence 'which is dependent also on the installation procedures of the soil around the sensors' as follows 'which is dependent also on the installation procedures of the sensors into the soil'.

Line 348: 'Extraction' instead of 'exctraction'.

Line 348: I suggest to change the sentence 'and this in a different way in the different seasons' by ', which varies throughout the seasons'. Consider to re-write it.

Line 349: Consider to remove 'always quite'. It sounds too subjective.

Line 361: 'LO' instead of 'LP'.

**Figures and Tables:**

Figure 1: Indicate the measurement unit for s (i.e., kPa).

Figure 2: Define Figs. 2a for the diagram and 2b for the upper-right inset in this figure.

Figure 3: Add coordinate grid, North arrow, and scale bar in Fig. 3a. Also indicate the source of the topographic map in Fig.3a and add an inset that shows the location of the study site in the context of Italy.

Table 1: Explain the variables in this table along the manuscript or in the table caption.

Table 3: The word 'curve' is duplicate.

---

## Author Comment (AC4)

Table. Main characteristics of the installed intruments.

| ELETRONIC SENSORS | COMPANY | MODEL | MEASUREMENT RANGE | TEMPERATURE RANGE | ACCURACY |
|---|---|---|---|---|---|
| Data logger and data Acquisition System | Campbell Scientific Inc. (Logan, UT, USA) | CR-1000 | - | -25°C to +50°C | ±(0.06% of reading +offset) at 0 °C to 40 °C for Analog Voltage |
| Time Domain Reflectometer | Campbell Scientific Inc. (Logan, UT, USA) | TDR-100 | -2 m to 2100 m (distance) and 0 to 7μs (time) | -40°C to +55°C | - |
| Multiplexer for TDR System | Campbell Scientific Inc. (Logan, UT, USA) | SDM8 x 50 | 8-Channel | -40°C to +55°C | - |
| Tension Transducer | Soil Moisture Equipment Corp. (Goleta, CA, USA) | 5301 | 0–85 cBar | 0°C to +60°C | 0.25% |
| Rain gauge | Campbell Scientific Inc. (Logan, UT, USA) | ARG100 | 0–500 mm/hr | - | - |

---

## Author Response (AR1)

**REPLY TO TOPICAL EDITOR**

Dear Topical Editor,
we took into account all issues raised by the Reviewers. Particularly regarding the main modifications provided, please note that:
- the Abstract was changed;
- the section 2 was reorganized by inserting a "Data and methods" section;
- the "Discussion" section was expanded;
- an aerial photo of the investigated site was added (Fig. 3b);
- a Figure showing a scheme of the installed instruments was added (Fig. 7);
- a specific table reporting the main features of the installed instruments was added (Table 3);
- six figures were modified (previously named Fig. 3a; Fig. 3b; Fig. 7; Fig. 8; Fig. 9; Fig. 10).

Please find listed below:
- reply to Referee #1 (from page 2 to page 4);
- reply to Referee #2 (from page 5 to page 8);
- reply to Referee #3 (from page 9 to page 15);
- reply to Referee #4 (from page 16 to page 24);

All the modifications will be referred to the marked-up version of the manuscript, that contains the updated versions of all the figures.

Best regards.
Luca  Comegna, on behalf of all the authors.

**OVERVIEW COMMENTS BY REFEREE #1**
**This research paper presents a dataset of hydrological measurements carried out from 2011 to 2012 in a shallow deposit of loose pyroclastic soils located in Campania, Southern Italy.**
**The work is well written and illustrated with figures and tables, and their conclusions seem to be consistent with the results and discussion. However, the restructuring of some parts of the paper is considered appropriate (see 'line-by-line commentary'). Authors included data and methods in the 'Monitoring site' section. An independent and clear section for 'Data and methods' may be very appropriate, especially in a data description paper like this.**
**General vote: manuscript may become acceptable after major changes, which are detailed in this review.**

**Reply by Authors**
Dear Referee #1,
many thanks for your suggestions that will undoubtedly improve the quality of the manuscript. In particular, we re-organized the work for the sake of clarity, including an appropriate "Data and methods" section.

The replies to the specific comments are reported in the follow. Please, consider that all the modifications will be referred to the lines of the marked-up version of the manuscript.

**LINE-BY-LINE COMMENTARY**
**R1 - Line 10: I suggest to change 'had been involved' by 'was involved'.**
A1 – This sentence was eliminated in the modified version of the Abstract (line 17).

**R2 - Lines 18: (e.g., Mualem, 1976…).**
A2 – We added "e.g." before the list of citations (line 26).

**R3 - Lines 22: 'specimen' instead of 'sample'?**
A3 – We provided the suggested change (line 33).

**R4 - Lines 30, 143, 150, 159, 207, 254, 278, 351, Figs. 8 and 10: Please, use the same style of English spelling along the manuscript (American or British English spelling). In some cases, authors have used British English spelling.**
A4 – You are right. We provided the suggested corrections using the same British English spelling. In particular, the word "characterized" was systematically replaced by "characterised". The word "hypothesized" was eliminated in Fig. 6 (previously named Fig. 5), Fig. 10 (previous Fig. 8) and Fig. 12 (previous Fig. 10).

**R5 - Line 52: Remove 'fully'?**
A5 – The sentence originally containing "fully" was removed (line 83).

**R6 - Line 55: 'was' instead of 'had been'.**
A6 – That sentence was removed (line 86).

**R7- Line 77: (e.g., Fiorillo et al., 2001...).**
A7 – We added "e.g." before the list of citations (line 103).

**R8 - Line 92: I propose to define a 'Data and methods' section in order to describe data and methods relative to the analysis of the monitored deposit from 2011 to 2012, whose results are presented in this work.**

A8 – We reorganized previous Section 2, defining it as a "Data and methods" section (line 91) in order to help the reader understanding the analysis of the field data described in the following Section 3 ("Results of field monitoring"). In particular, this section 2 contains the following subsections:

- 2.1 Geomorphological and climate framework (line 92);
- 2.2 Soil properties (line 138);
- 2.3 Monitoring station (line 209).

Moreover, a specific Table 3 (line 245) reporting the main characteristics of the installed instruments was added.

Such section also contained the Equations 2 (line 253) and 3 (line 260) used for the slope stability analyses discussed in the "Discussion" section (Section 4).

**R9 - Line 119: Consider to put this paragraph at the end of the 'Data and methods' section.**

A9 – Such paragraph was changed and put at the end of the "Data and methods" section (line 265).

**R10 - Line 131: Please, explain what is each variable in the manuscript.**

A10 – The meaning of the reported variables was explained in the text (line 193).

**R11 - Line 194: Remove 'further and further'?**

A11 – "growing further and further" was replaced by "flourishing" (line 333).

**R12 - Line 342: Consider to re-write the sentence 'which is dependent also on the installation procedures of the soil around the sensors' as follows 'which is dependent also on the installation procedures of the sensors into the soil'.**

A12 – The suggested correction was provided (line 515).

**R13 - Line 348: 'Extraction' instead of 'exctraction'.**

A13 – The typo was fixed (line 521).

**R14 - Line 348: I suggest to change the sentence 'and this in a different way in the different seasons' by ', which varies throughout the seasons'. Consider to re-write it.**

A14 – The suggested modification was provided (line 521).

**R15 - Line 349: Consider to remove 'always quite'. It sounds too subjective.**

A15 – We agree and we removed "always quite" (line 523).

**R16 - Line 361: 'LO' instead of 'LP'.**

A16 – "LP" is correctly reported (line 553).

**FIGURES AND TABLES**

**R17 - Figure 1: Indicate the measurement unit for s (i.e., kPa).**

A17 – As the represented curves are qualitative (i.e. without reference to any real soil or experimental data), we think it is not worth to insert units here (which instead are reported in Figure 2).

**R18 - Figure 2: Define Figs. 2a for the diagram and 2b for the upper-right inset in this figure.**

A18 – The suggested modifications were provided.

**R19 - Figure 3: Add coordinate grid, North arrow, and scale bar in Fig. 3a. Also indicate the source of the topographic map in Fig.3a and add an inset that shows the location of the study site in the context of Italy.**
A19 – Figure 3a was changed. In particular, we inserted: coordinate grid; North arrow; scale bar; the location of the site reported on the map of Italy (including the geographic coordinates). As specified in the caption, the topographic map was provided by Damiano, 2004.

**R20 - Table 1: Explain the variables in this table along the manuscript or in the table caption.**
A20 – The meanings of the reported variables were explained in the table caption (line 150).

**R21 - Table 3: The word 'curve' is duplicate.**
A21 – The typo was fixed (line 344).

**OVERVIEW COMMENTS**
**Comment by Referee #2**
The work addresses a remarkably interesting topic. The characterization of hydraulic hysteresis in situ and the role of such dynamics on slope stability can provide useful information for landslide practitioners. In this regard, the work represents a valuable contribution to research on such topics. According to my view, it could be suitable for publication after minor revisions are implemented.

**Reply by Authors**
Dear Referee #2,
many thanks for your comment. We really appreciated your positive assessment. We sincerely hope that the paper, which highlights the crucial role of field monitoring to understand the slope response to weather forcing, can provide useful information to landslide experts.
The replies to the specific comments are reported in the follow.

**LINE-BY-LINE COMMENTARY**
**R1 - ABSTRACT: In my view, the Abstract should be reviewed. First the main Target could be introduced, after the collected data to investigate the issue could be presented. In this regard, they should be differentiated: volumetric water content and suctions are used to investigate the soil hysteretic behaviour while the other ones provide the boundary conditions regulate the exchanges among soil, vegetation, and atmosphere. Furthermore, temperature data could be cited also if not directly managed by the Authors.**

A1 – The Abstract was modified following the provided suggestions.

**R2 - In my view, the significant detail about the role of imbibition/drying rate deserves more attention in the initial description. In this regard, it should be presented in advance with few sentences about all the mechanisms regulating the hysteretic process (air entrapment, ink effect). Rows 46-51 could be anticipated.**

A2 – As suggested, rows 46-51 were anticipated in order to present in advance the hysteretic mechanism (lines 29-33).

**R3 - Figure 2: please introduce T and TDR in the caption of Figure 2.**

A3 – In order to avoid confusion, the letter "T" (originally shown by the previous figure) was replaced in the current Figure 2b by the word "tensiometer".

**R4 - L52: the investigation of soil hysteretic behaviour is usually carried out by using laboratory tests. Which are the main benefits associated to the use of field data (also accounting for the higher difficulties in retrieving and post-processing data)?**

A4 – The laboratory tests are, of course, of prominent importance because they allow to precisely define the soil response to imposed well known boundary conditions. Field data however may help, since they can

highlight different and sometimes unexpected factors which also affect the hydrologic response of natural slopes and that can be recognized and understood only by in-situ monitoring.

**R5 - L57-58: temperature data are also available.**

A5 – We provided such change in the text (line 64).

**R6 - Figure 4: which are the reference time spans over which the average values are computed? How long is the interval? If possible, a measure related to the "spread" could provide information about interannual variability.**

A6 - The monthly mean temperature data used for the estimation of PET are calculated with the 1979-1998 monthly temperature data from the meteorological station of Montesarchio, managed by the National Hydrologic Service, which was located some 4 km from the test site, approximately at the same elevation (560 m a.s.l.). The minimum and maximum mean monthly temperatures are for all months within ±2.3°, corresponding to maximum variations of few mm per month of the estimated PET. These small variations, which typically compensate throughout single years, do not significantly affect the slow water exchanges due to effective evapotranspiration. Rainfall data used for the calculation of the mean monthly rainfall are from the rain gauge of Cervinara, belonging to the meteorological alert network managed by the "Functional Centre for forecast, prevention and monitoring of risks and alerting for civil protection" of Campania Civil Protection Agency. Rainfall variability is obviously higher, compared to temperature, but the aim of Figure 4 is to give information about the local climate, and we think that inserting minimum and maximum monthly values would make the graph less readable without giving clear information to the reader (monthly precipitation deviations from the mean also usually compensate throughout the year).

Some information were provided in the text (lines 125-131).

**R7 - L119: please provide brief insights about the choice of such period. For example, a comparison between temperature, rainfall heights, PET of the selected period and the average trends could be useful to retrieve its peculiarities joining the content of Figure 2 and 6.**

A7 – Such period has been chosen because of the abundance of available data about both suction and moisture content. We provided a short insight to help the comparison between the average (Figure 4) and specific (Figure 8) weather data (line 274; lines 280-287).

**R8 - Figure 5: please report the paper(s) where the meaning of the different acronyms used in the legenda can be retrieved. If relevant for such investigation, please report also in the Manuscript.**

A8 – The acronyms in Figure 5 (now Figure 6) are those reported in the provided repository dataset. We clarified that in the caption (line 201-202).

**R9 - Table 2: please replace "q" with "theta"; for theta, please use the same symbol.**

A9 – Thanks for your advice. It's likely that the format has been accidentally changed. Of course, we provided the requested correction (line 206).

**R10 - L157: please mention that, as the station is not located on the slope and probably at different altitude, some discrepancies can arise. If possible, include the position of temperature sensor in Figure 3 or provide some details about the location.**

A10 – We provided such consideration in the text (line 287-288).

**R11 - L191: please provide more details about iso safety factors permitting a clearer understanding about their meaning (e.g. anticipating lines 301-312).**

A11 – We anticipated the considerations about the factor of safety, FS, within subsection 2.3 of "Data and methods" in order to help the understanding of the contents provided by Section 4 (lines 250-263).

**R12 - L196: please consider that, probably, in the final part of the period, the atmospheric demand could be not fulfilled by soil water.**

A12 – We agree. Indeed, we did not mean that actual evapotranspiration is in any case able to totally accommodate PET, but just that during the period of flourishing vegetation it may approach this limit. We modified the sentence and added the suggested considerations (lines 335-336).

**R13 - L196: please introduce further details about how main drying curve has been identified.**

A13 – As explained in lines 340-341 and 352-354, the two man drying curves have been identified by fitting with the Equation (1), proposed by van Genuchten (1980), the experimental points monitored during the driest time windows (red curves in Figure 10a and 10b). We assumed the same saturated and residual volumetric water contents ($\theta s = 0.75$; $\theta r = 0$) already assigned to the main wetting curve (determined by lab tests shown in Figure 2) and $m = (n\text{-}1)/n$ (according to Mualem, 1976), thus identifying only the parameters $\alpha$ and $n$.

**R14 - L261: please provide, if possible, an attempt value for field capacity suction for your soil.**

A14 – As reported in line 327, the field capacity of investigated soil is about 0.25 $m^3/m^3$. We reminded such value in line 418 too.

**R15 - L268: probably, internal hysteresis within the scanning curves could be not relevant (the same insight can be retrieved from other Research papers).**

A15 – The internal hysteresis is not particularly relevant according to our field data too. However, some little differences between the drying and the wetting scanning paths have been recognized at both depths when the highest measured water content is attained (next to the main wetting curve). In particular:

- at the depth z = 0.60 m, curve FG is above curve EF (Fig. 12a)
- at the depth z = 1.00 m, curve HM is above curve EH (Fig. 12b)

We remarked such observation in Section 4 (lines 458-463).

**R16 - L314-317 If possible, I suggest also including the trends in terms of safety factors in Figure 7 and 9 (Figure 11 and 12 are then anticipated).**

A16 – As suggested, we eliminated previously named Figure 11 and 12, inserting the corresponding safety factor trends in Figure 9d and 11d.

**OVERVIEW COMMENTS**

**Comment by Referee #3**

Comegna et al. present an interesting dataset of soil moisture and suction and associated relationships over approximately one hydrologic year. As such, the manuscript is an interesting read and the data likely have utility for modelling. That said, the authors do not offer much in the way of suggesting potential uses of the data unless I just missed that somehow. There are also many issues to be addressed in revision. I provide specific comments below for consideration in revisions. In revision, I suggest the authors clearly address the value of the dataset beyond the presented use. Limitations should also be discussed.

**Reply by Authors**

Dear Referee #3,

many thanks for your comments, that will be very useful to improve the quality of the manuscript. Even though highlighted by many field and laboratory evidences (reported in literature and acknowledged in the manuscript), the hydraulic hysteresis of unsaturated soils is still often neglected by researchers involved in modelling the hydrological slope response to weather forcing for landslide prediction, who usually adopt a single Soil Water Retention Curve fitting all the available experimental data. Such choice is mostly made because the adoption of a more sophisticated retention model would require information about specific wetting/drying paths at different initial conditions that are frequently not available. The proposed paper is therefore aimed to provide data (monitored in-situ at a detailed time scale) on this point. We tried to better stress this point in the Abstract and in the Introduction. The replies to the specific comments are reported herein after.

**SPECIFIC COMMENTS**

**R1 - Abstract, Line 10: Consider changing "concern" to "include".**

A1 – The original sentence containing that word was eliminated in the modified version of the Abstract (line 17).

**R2 - Abstract, Line 10: Rainfall is referred to as a height here and throughout the manuscript. Consider referring to it as a depth and change throughout.**

A2 – Rainfall was referred to as a depth here (line 14) and throughout the text.

**R3 - Abstract, Lines 11-13: This entire sentence needs revision for clarity. For example, "...the installation at the same depths..." refers to installation of what specifically?**

A3 – We modified the sentence (lines 18-19), specifying the installed instruments (tensiometers and TDR sensors).

**R4 - Abstract: In general, the abstract is not particularly informative and should be revised to more clearly explain what is being provided and of what potential use(s) that content serves.**

A4 – The abstract was changed in order to better clarify the meaning of the provided information and their potential use.

**R5 - Page 1, Line 27: The text "that prevents to reach" needs revision for clarity. The meaning isn't clear to the reader.**

A5 – The sentence was replaced by "that does not allow full soil saturation" (lines 38-39).

**R6 - Page 1, Line 29, The text "If moving along one of these paths a reverse process is initiated" is awkward.**

A6 – The sentence was replaced by "If a reverse process takes place along one of these paths" (line 40).

**R7 - Page 2, Lines 33-34: What is the point of this text?**

A7 – The sentence "Concerning this point, Tami et al. (2004) report the results of some tests…" will be replaced by "This result has been experimentally recognized by Tami et al. (2004), through some tests…" (line 45).

**R8 - Page 2, Line 36: TDR should be defined here.**

A8 – TDR acronym was defined here (line 48).

**R9 - Page 4, Figure 3: The drawing of the soil moisture probes is misleading. From the text, they are either 10 or 40 cm long. 40 cm would be much longer than what is drawn. Also, Figure 3A is too small. It's hard to read.**

A9 – Figure 3a was enlarged and modified. The modified version of Figure 3b (now Figure 5) shows the soil moisture probes properly scaled accounting for their actual size.

**R10 - Page 4, first line on the page: Consider changing "…concern the time period going…" to"…concern the time period from…"**

A10 – That sentence was eliminated (line 88).

**R11 - Page 4, Line 60: The word "far" is not necessary here.**

A11 – The word "far" was eliminated (line 94).

**R12 - Page 4, Line 62: "…precipitations…" should be "…precipitation…"**

A12 – The word "precipitations" was replaced by "precipitation" here (line 128) and throughout the text.

**R13 - Page 4, Line 62: Insert "by" before "warm and dry summer" to read, "…and by warm and dry summers."**

A13 – That sentence was eliminated in the updated text (line 98).

**R14 - Page 5, Line 80: What is meant by "…some layers locally miss…"**

A14 – The sentence "Some layers locally miss" was replaced by "In some verticals, some layers were not found" (line 106).

**R15 - Page 5, Line 83: Can the text "get dry and fall" be changed to "…Leaf senescence occurs in October…" or something like that.**

A15 – The text was simplified to "In October, leaves fall from the trees…" (line 110).

**R16 - Page 5, Lines 86-88: This is a really long sentence and should be broken up a bit for clarity.**

A16 – The original long sentence was broken into three sentences (lines 112-116).

**R17 - Page 6, Line 92: What is "altered ash"?**

A17 – The expression "altered ash" is used to indicate a deteriorated ash layer with a grain size which is turning from silty sand to clayey and silty sand. We clarified it in the text (lines 144-146).

**R18 - Page 6, Table 1: Need to define columns. Table should be able to be interpreted independent of text. Consider this for all tables in the manuscript.**

A18 – We clarified the used symbols in the captions of all the tables, in order to make their interpretation independent of text.

**R19 - Page 6, Lines 94-97: Need to describe methods for determining these soil characteristics.**

A19 – The reported soil properties have been provided by Damiano et al. (2012) through a number of field investigations and geotechnical laboratory tests (carried out both on undisturbed and reconstituted soil samples). We mentioned that in the text (lines 139-142).

**R20 - Page 6, Line 100: How many paired probes in total? What was the spatial sampling design? A table describing the location of all of your probes would be very helpful. And, introduction says data pertain to 2011 and 2012. Are these different data?**

A20 – We understood that probably too emphasis was given by the old version of the text to the previous monitoring activities described by Damiano et al. (2012), so that the reader might have expected the corresponding data to be discussed. Therefore, in order to avoid confusion, we eliminated that part (lines 159-164).

Concerning the data described in the proposed paper, two TDR probes have been installed in a row with two tensiometers and at a distance of 30 cm from them. Their centre is at a depth very close to the centre of the ceramic tips of tensiometers. This information was added in the text and was brought to evidence in the modified version of Figure 3b (now Figure 5). Moreover, we added a scheme of the installation of the instruments described in the paper (Figure 7).

The described monitoring data are available from January, 2011, to January, 2012 (13 months). We clarified that in the text (lines 65; 265).

**R21 - Page 6, Line 102: What kind a rain gauge? More detail needed on this and the "Jet fill" instruments, other instruments as well. Manufacturer, etc.**

A21 – A specific Table 3, reporting the main characteristics of all the installed instruments, was inserted.

**R22 - Page 6, Line 104: Should "…has been installed…" be "…was installed…"**

A22 – We provided the suggested modification (line 210).

**R23 - Page 6, Line 105: 7 TDR probes or 7 metallic rods?**

A23 – That sentence was eliminated (line 163).

**R24 - Page 6, Line 111: 100 or 400 mm? That's a very large area for TDR measurements. More detail needed? Is this the same sampling area as the tensiometers?**

A24 – The two TDR probes described in the manuscript are 40 cm long, while the tensiometer ceramic tips are 7 cm long. We outlined in the Discussion section that the suction and soil moisture transducers sample different soil volumes that might contribute to imperfect matching of suction and moisture data (lines 465-466).

**R25 - Page 6, Line 119: That is not what the following section describes. The structure of the paper is difficult to follow. Maybe a header to describe lab experiments here?**

A25 – We reorganized the text, inserting a "Data and methods" section (Section 2) in order to help the reader understanding the analysis of the field data described in the following Section 3 ("Results of field monitoring"). In particular, Section 2 was organized through the following subsections:
- 2.1 Geomorphological and climate framework (line 92);
- 2.2 Soil properties (line 138);
- 2.3 Monitoring station (line 209).

Results of lab experiments are reported in subsection 2.2 "Soil properties".

**R26 - Page 6, Line 121: 40 or 10 cm? Does z equal the center or the bottom of the tensiometers and TDR probes?**

A25 – The centre of the tensiometer ceramic tip is located at the same depth as the centre of the TDR probe (lines 226-227).

**R27 - Page 6, Line 122: How might the different volumes influence the results?**

A27 – Due to the different soil volumes tested by tensiometers and TDR probes, any variation in suction and in water content variations is not simultaneously detected by the sensors. For instance, if the wetting front advances downward, the 40 cm long TDR probe can detect it with some advance compared to the ceramic tip of the tensiometers (the centres of the two sensors are aligned, so the upper edge of the TDR probe is some 16 cm above the upper edge of the tip). The temporal mismatch will be larger for steeper wetting front. A similar issue would affect the edges of the sensors during vertical (gravity-driven) drainage processes, while soil drying caused by root uptake is expected to be more evenly distributed throughout the entire root zone. These issues would certainly affect the coupling especially in the initial stage of infiltration and drainage process, which are characterized by steeper gradients. However, the measurements are acquired every six hours, and looking at figures 7 and 9, it appears that suction and water content variations at the two investigated depths, 40 cm apart from each other, are detected nearly simultaneously. Hence, it is expected that the temporal mismatch may affect only one or two measurement points during each of the wetting/drying paths discussed in figures 8 and 10, which consist of many measurement points as they refer to long lasting processes.

Discussion section was expanded in order to insert these considerations (lines 465-476).

**R28 - Page 7, 123: How does this section relate to the field data collected? It would be good to be explicit here. Are these data included in your repository on Zenodo?**

A28 – These lab results, included in our repository on Zenodo (lines 548-549) allowed to hypothesise in the retention plane a reliable lowest boundary (i.e. the main wetting curve) for the field data. We explicitly comment this goal at lines 196-198.

**R29 - Page 7, Table 2 caption: What is "lowest water retention boundary" mean?**

A29 – Through this expression we intended to indicate the "main wetting curve" that represents the lowest boundary of the hysteresis domain in the retention plane, i.e. we assume that all the experimental points fall above or along this curve (never below it).

In order to avoid confusion, we decided to use the definition "main wetting curve" in the updated text (line 204).

**R30 - Page 8, Line 142: What do you mean by "cover"? Be more specific…litter cover, basal plantcover…?**

A30 – We wanted to indicate the whole "pyroclastic cover". Anyway, that sentence was eliminated (line 270).

**R31 - Page 8, Lines 144-146: More detail on the local weather station is needed, since PET is referenced frequently.**

A31 – These temperature data (shown in Figures 8, 9a and 11a) are provided at a hourly scale since 2002 by the Pietrastornina weather station (located at 495 m a.s.l. and 15 km from Cervinara) and managed by the "Functional Centre for forecast, prevention and monitoring of risks and alerting for civil protection" of Campania Civil Protection Agency.

The mean monthly temperature used for the estimation of PET (Figure 4) have been instead calculated with the 1979-1998 monthly temperature data provided by the meteorological station of Montesarchio, managed by the National Hydrologic Service, located some 4 km from the test site at the elevation of 560 m. Such station does not provide hourly data.

Both the meteorological stations are very close to the considered slope and are located at approximately the same altitude as the field monitoring station. We added all this information in the revised manuscript (lines 129-131; 235-237)

**R32 - Page 8, Lines 160-162: This section needs some revision for clarity and to point out "missing data" rather than "data lacks". Also, the period of missing data is nearly half of the time you are highlighting. Why did you choose these two probes and this time period if there is such a large data gap?**

A32 – The period with no data ("missing data" that will replace "data lacks" in the text) concerns about 35% of the monitoring period (going from January, $1^{st}$, 2011 to January, $31^{st}$, 2012). Such period has been chosen because, despite the declared gap, it allowed to recognize important aspects of the hysteretic hydrologic response of the investigated deposit. We added this consideration (line 299-300).

**R33 - Page 9, Section 3.1: Should provide an overview sentence here or table that explains that A-H are referring to specific dates (e.g. A = January 1st).**

A33 – The dates associated to the different letters were added in the legends of Figure 10 and Figure 12.

**R34 - Page 9: For the discussion on the various windows of time and associated trends, I think a graph of cumulative precipitation, ET, and soil moisture would be beneficial to the reader. Those allow the user to more easily see trends in wetting and drying in relation to water inputs (precip) and water losses (ET). I know previous similar studies have even looked at water input – evapotranspiration as a useful metric.**

A34 – We see Reviewer's point: the assessment of the water balance of the soil profile and, more specifically, of the investigated layers, would be useful for a more complete interpretation of the flow processes behind the observed hysteretic behavior. However, there are several reasons for which this kind of assessment (and the corresponding suggested graph) is not suitable in this case. First of all, there wasn't a complete meteorological station operating at the slope, so it is not possible to evaluate ET (we have only mean PET). Second, the water balance should be completed also considering the leakage towards the underlying part of the soil profile. Third, the depth at which the shallowest sensors were installed (60 cm below the ground surface) implies that part of the infiltrating water could be subtracted from the water balance at the considered depths because of possible (undetectable) variations of water content in the uppermost part of the soil profile. However, the discussed hysteretic behavior is detected by means of local measurements, and the observations remain valid also without information about what happened above and below the considered depths.

**R35 - Page 9, Line 184: Units for "-1"?**

A35 – That sentence was completely modified (lines 321-323).

**R36 - Page 9, Line 186: Do you mean infiltration? If so, how about just saying that.**

A36 – Indeed, we are discussing the effects of an infiltration process, but speaking simply about "infiltration" could mislead the reader, as it would not underline that the investigated soil layer receives water from the top but at the same time also releases water towards the underlying part of the soil profile (so, what we see is the result of the (un)balance between infiltration and drainage).

**R37 - Page 9, Line 188: How is this threshold value determined?**

A37 – The field capacity is conventionally defined as the value of water content corresponding to a suction between10 kPa and 33kPa after a gravitational drainage process. Looking at the experimental points of Figures 10 and 12, a value between 0.20 and 0.30 seems plausible, although the presence of intense root water uptake does not allow to clearly identify the effects of gravitational drainage alone.

**R38 - Page 9, Line 190: How is the curve fit?**

A38 – The plotted curve is just a manual fit. We remarked that in the text (line 329).

**R39 - Page 9, Line 194: The text "…starts growing further and further…" – what does this mean?**

A39 – The expression "growing further and further" was replaced by "flourishing" (line 333).

**R40 - Figures: For nearly all figures, the captions could be greatly improved to inform the viewer what is depicted. For example, the caption for the graphs in Figure 10 say nothing about what the different upper case letters represent. This same issue occurs in many of the captions. As another example, the inset figure for Figure 2 is not explained. Nearly all captions need substantial improvement for clarity.**

A40 –As suggested, the captions of all figures were improved to give more information to the reader.

**R41 - Conclusions: Are the highlighted data consistent with the rest of the data?**

A41 – The highlighted data are consistent with the rest of the data.

**R42 - Data Availability: The data structure is not described at all in the paper. The description is central to the utility of the data for future analyses and a core component of publishing in ESSD.**

A42 – The repository Zenodo includes field data monitored in the investigated period (rainfall, temperature, suction and soil moisture) and laboratory infiltration test (shown in Figure 6). Such data have been collected in different Excel files. A brief description of how such spread sheets are organized was added to the section "Data availability", at the end of the manuscript (line 535-549).

**OVERVIEW COMMENTS**

**Comment by Referee #4**

The manuscript presents field measurements of rainfall, volumetric soil moisture content and soil suction. Hysteretic wetting and drying paths are shown in the data. The manuscript is reasonably well written, an interesting data set is presented and it describes an interesting phenomenon in the data (hysteretic wetting and drying paths). The data set is well referenced in the manuscript and it is accessible. At the data repository the data are clear and complete, including metadata.

The manuscript nicely analyses hysteretic wetting and drying paths in a data set of volumetric soil moisture content and soil suction. However, I am not sure whether this manuscript's objective fits the aims and scope of the ESSD journal. According to the aims and scope formulated for the ESSD journal, the ESSD journal aims at furthering the reuse of data. Articles may pertain to the planning, instrumentation and collection of data. Interpretation is outside the scope of regular articles. Besides this point, improvements can be made to the description of the context and the methodology for analysing the hysteretic wetting/drying paths. These two points are reflected in the specific comments below: some comments are on more extensively describing the monitoring site and the data collection (specific comments 3, 4, 5, 6), whereas others are on the introduction, methodology and conclusion regarding the analysis of the wetting/drying paths (specific comments 1, 2, 7, 8, 9).

**Reply by Authors**

Dear Referee #4,

many thanks for your comments and suggestions, that give us the opportunity to clarify some aspects and that will be properly used to improve the quality of the manuscript.

Regarding the remarks about the contents of the paper and its ability to fit with the aim of ESSD, the Authors imagine that further reuse of data does not necessarily imply that no comment about them could be made by the presenters. By the way, such comments could work as a starting point for further analyses, which might even lead to a completely different perspective.

The replies to the specific comments are reported in the follow.

**SPECIFIC COMMENTS**

**R1 - The introduction does contain a nice description of the hysteretic nature of main wetting/drying paths and scanning curves, and Figure 1 is very illustrative. It also introduces a laboratory experiment on these phenomena. However, the research gap leading to the manuscript's research objective could be better defined in the introduction. For example, is this the first manuscript in which the hysteretic nature of wetting/drying paths are shown in field measurements? Or if that is not the case, what is unique to this data set and study area?**

A1– Even though often observed in laboratory experiments, the hydraulic hysteretic response of unsaturated soils is still often neglected at slope scale, being usually modelled by a single Soil Water Retention Curve. Such a choice is frequently due to unavailability of detailed field information. Most of the knowledge is in fact based on the results of laboratory investigations and/or of physical modelling. Such tests, although very useful, are unavoidably not able to take into account further aspects that that could make the actual hysteretic response of natural slopes quite different from what is observed in the lab such as the influence of different boundary conditions, the role of root water uptake or of the atmospheric evaporative demand, etc. The proposed manuscript is so aimed to give a contribution in this direction. We stressed such points in the modified version of the Introduction (lines 54-60).

**R2 - The topic of the manuscript does not seem well represented by the reference list. 17 out of the 34 items in the reference list are related to author(s) of the manuscript.**

A2 – Unfortunately, the literature is not rich on this subject. The best examples known by the authors are however reported in the reference list. The mentioned Authors' papers report data from laboratory and field investigations, or of physical and mathematical modelling concerning the same investigated site. We believe that these references can be useful to the readers to get more information about properties and behaviour of the same soils, and to know concerned stability problems that often affect this kind of slopes (which is the main reason why the monitoring campaign was carried out). Anyway, we eliminated four references related to authors of the manuscript without loss of information (lines 608-615; lines 618-620).

**R3 - The manuscript presents data for January 2011 to January 2012. Has the data collection in this area not been continued after January 2012? It is stated that the data collection in this area started already in 2002, so it would be nice to also describe the data of 2002 to 2011 in this manuscript (as submitted to the ESSD journal). References to other data sets for this area would also be useful, such as meteorological measurements and discharge measurements, as this helps future reuse of the presented data set by other research groups.**

A3 – We understood that probably too emphasis was given by the old version of the text to the previous monitoring activities (period 2002-2009) described by Damiano et al. (2012), so that the reader might have expected the corresponding data to be discussed in the paper and to be made available in the open repository. Therefore, in order to avoid confusion, we eliminated that part (lines 159-164).

The automatic monitoring station was installed at the end of 2009, but simultaneous information about soil moisture and suction at the same depth (necessary to highlight the hysteretic response) are available only since 2011. Due to maintenance and integration of the instrumentation, the automatic monitoring had to be stopped after January 2012 and it has been restarted only recently.

Regarding the meteorological data, the repository on Zenodo contains data about rainfall and temperature hourly values. Since November 2017 monitoring data at the same slope are again being collected, including more meteorological variables and also discharge measurements in nearby streams. Some of them are reported in Marino et al. (2020), which is acknowledged in the reference list, and is then accessible to the readers.

**R4 - Line 62-64 and Figure 4: are these values the climatology values for rainfall, temperature and potential evapotranspiration? Over which period of years were these values calculated? Furthermore, are these values estimated for the monitoring site or based on measurements at a nearby meteorological station?**

A4 - The monthly mean temperature data used for the estimation of PET are calculated with the 1979-1998 monthly temperature data from the meteorological station of Montesarchio, managed by the National Hydrologic Service, which was located some 4 km from the test site, approximately at the same elevation (560 m). Rainfall data used for the calculation of mean monthly rainfall are from the rain gauge of Cervinara, belonging to the meteorological alert network managed by the Regional Civil Protection, some 2 km far from the monitored slope. We added information about the source of climatological data in the revised manuscript (lines 125-131).

**R5 - The monitoring site should be described in more detail. From Line 101-118, which describes the set-up of the monitoring site, it is not clear at which depth the Jet-fill tensiometers of 2002, the Jet-fill tensiometers of 2009 and the TDRs are installed. Could you make a schematic cross section of all the sensors (including types, such as 40 cm and 10 cm TDRs) and at which location and depth they were installed? Fig. 3b only shows a few sensors. Furthermore, what are other characteristics of the monitoring site? For example, is the location uphill or downhill and how does this affect the measurements? It would also be interesting to see photos of the monitoring site, showing i.a. the position of logger(s) and sensors, land cover and surroundings.**

A5 – As anticipated in reply A3, monitoring data from 2002 to 2009 provided only rainfall and suction data (measured manually) already published by Damiano et al. 2012. We eliminated that part from the text to avoid misleading.

Data described in the proposed manuscript concern only those automatically recorded by the two couples of described instruments installed at very close depths, thus able to highlight the hysteretic soil response. The TDR probes have been installed in a row with the two tensiometers and at a distance of 30 cm from them (Fig. 4a). The four instruments are about 5 m far from the data logger. As suggested, we reported such additional information in the text (line 225) and in Figure 5, adding also some photos (Fig. 7).

**R6 - It is very good that the authors mention the sensitivity of the rain gauge and the calibration accuracy of the TDR. What is the uncertainty of the tensiometers? What is the brand and type of the rain gauge?**

A6 – A specific Table 3, reporting the main characteristics of the installed instruments, was inserted.

**R7 - Line 123-135 and Table 2, 3, 4: The methodology of fitting the Van Genuchten parameters was not described in sufficient detail. Were the measurement pairs from all nine experiments used? Which algorithm was used for the fitting? Was the fitting done specifically for this manuscript or was it done previously for another paper and did you re-use their result?**

A7 – The three van Genuchten curves are featured by the same saturated and residual volumetric water contents ($\theta s$ = 0.75; $\theta r$ = 0), both assigned according to previous experimental results on undisturbed specimens (e.g. Damiano et al., 2012). According to Mualem (1976), the parameter m has been related to the parameter n by the equation m = n-1/n. Therefore, two are the remaining parameters to be fitted: $\alpha$ and n.

Regarding the assumed lowest boundary (main wetting curve), the chosen $\alpha$ and n parameters (shown in Table 2) can provide the best fitting of all the experimental data located along the steepest part of each infiltration test. As added in text (line 189) such points correspond to suction values smaller than 4 kPa.

Concerning the assumed upper boundaries, the $\alpha$ and n parameters have been chosen in order to provide the best fitting of the experimental points monitored during the time windows featured by the highest observed evapotranspiration effects, namely:

- at the depth z = 0.60 m, from May 8[th] (point B in Figure 10a) to July 18[th] (point D in Figure 10a)
- at the depth z = 1.00 m, from June 22[nd] (point C in Figure 10b) to July 18[th] (point D in Figure 10b)

**R8 - The last part of the conclusion is rather vaguely formulated. Do the results raise questions or do they allow to draw conclusion from the results on how we should set up reliable early warning systems? Please also check the last sentence of the conclusion (Line 354-355).**

A8 – The results of this study should first of all highlight the limited reliability of early warning systems that use empirical threshold models exclusively based on the characteristics of precipitation (typically intensity and duration) without accounting for the prominent role played by the hydraulic properties and the initial state of the soil (that is also related to vegetation). Therefore, a reliable early warning system should be based on:
- a monitoring system able to provide real time updates about the weather-induced hydraulic paths;
- a forecasting model accounting for the soil water retention properties.
Both these aspects could be supported by the knowledge of the main drying and wetting curves that bound the water retention domain.
All these observations were added in the revised Conclusions section (lines 528-531). The sentence at lines 354-355 (now 532-533) was removed.

**R9 - Section 2 is long and contains a variety of topics. Consider splitting it in a Section 2 ('Monitoring site') and a Section 3 ('Methodology'), or in subsections with headings that reveal what we can expect.**

A9 - We inserted a "Data and methods" section (Section 2) in order to help the reader understanding the analysis of the field data described in the following Section 3 ("Results of field monitoring"). In particular, Section 2 was organized through the following subsections:

- 2.1 Geomorphological and climate framework (line 92);
- 2.2 Soil properties (line 138);
- 2.3 Monitoring station (line 209).

Such section also contained the equations used for the slope stability analyses reported in Section 4 ("Discussion").

**R10 - In general the manuscript is well written, but the writing also needs some improvement. Consistency in terminology would increase the readability. For example, use one term for 'soil moisture content', 'soil moisture', 'volumetric water content' and 'θ' if you mean the same phenomenon. Next to that, if a symbol has been defined in the text, the symbol should be used from that point on and not interchanged with the term. Besides the technical corrections below, the manuscript would benefit from a check on the English writing.**

A10 – We really appreciate you considered the manuscript as well written. We followed your suggestions aimed at improving the consistency of terminology and at avoiding repeating the definition of variables instead of simply using the symbols. We also checked the English language of the entire manuscript.

**TECHNICAL CORRECTIONS**

**R11 - The unit of soil moisture content values is missing throughout the manuscript.**

A11 – We used the symbol $[m^3/m^3]$ throughout the revised manuscript.

**R12 - Line 36: Define what 'TDR' stands for.**

A12–We pointed out in the modified text that the acronym TDR stands for "Time Domain Reflectometer" (line 48).

**R13 - Figure 2: Define 'T' and 'TDR'.**

A13 – In order to avoid confusion, the letter "T" (originally shown by the previous figure) was replaced in the current Figure 2b by the word "tensiometer".

**R14 - Figure 3a: The topography of the study site is not clear from this figure. The figure also misses cartographic elements such as a north arrow and a scale (bar).**

A14 – Figure 3a was changed. In particular, it was enlarged and we inserted: coordinate grid; North arrow; scale bar; the location of the site reported on the map of Italy (including the geographic coordinates).

**R15 - Figure 8 and Figure 10: I suggest to use different colours for the different time windows so that readers can distinguish them when they are (partly) overlapping. This especially applies to Figure 10.**

A15 – The updated versions of Figure 8 (now Figure 10) and Figure 10 (now Figure 12) followed your suggestion.

**R16 - Table 1: How were these properties determined or what is the source? Furthermore, the symbols used in the table must be defined in the caption or, if they are not used at other places in the manuscript, they could just be written out.**

A16 – The soil properties, provided by Damiano et al. (2012), are the results of field investigations and of laboratory tests performed on several undisturbed and reconstituted soil samples. We pointed out such reference (line 140), indicating the symbols of the properties in the caption of Table 1.

**R17 - Line 110: I do not understand what you want to say with "… after cold periods when low temperatures could freeze." It should also be specified what you 'check' for in those periods.**

A17 – During the coldest periods, featured by temperature lower than 0°C, the de-aired water contained in the upper part of the hydraulic circuit of the tensiometers could freeze, thus affecting the correctness of the pressure transducer reading. If it occurred, we made note of that in order to give a correct interpretation of the corresponding registered data. Moreover, the expanded volume of ice could break either the pressure transducer or the plexiglass tensiometer tubes. If it occurred, they had to be fixed. These potential inconveniences were mentioned in the modified text (lines 219-223).

**R18 - Line 126: Should 'the field porosity' be changed to 'a porosity'? At Line 96 you describe that the porosity at the monitoring site ranges between 50% and 75%, so this is not 'the field porosity'.**

A18 –The model slope was formed with the Cervinara ashes, whose porosity ranges from 68 % to 75 %, as reported in Table 1. Therefore, we pointed out in the text that it had been reconstituted at the "maximum field porosity" (line 184).

Line 147 refers to the full range of field porosity, including also the pumices that, as reported in Table 1, present lower values (50-55 %).

**R19 - Line 130: Should 'steepest' not be 'steep'? I think you did not just fit to the steepest of the nine curves?**

A19–As previously explained in reply A7, the experimental points featured by a suction lower than 4 kPa have been fitted. We modified the text (lines 188-190).

**R20 - Line 134-135: Does 'possible reference lowest boundary' refer back to the main wetting curve and lowest limits as explained in the introduction? If that is the case, please use the same terminology. The same is the case between 'possible reference lowest boundary' in Line 134-135 and 'hypothesized lowest boundary' in Fig. 5.**

A20 – The "reference lowest boundary" coincides with the "main wetting curve" explained in the introduction. We used such less demanding expression because it has been derived by fitting a limited number of experimental points (those featured by suction lower than 4 kPa, as explained in A7). Anyway, in order to avoid confusion, we changed the text referring to the term "main wetting curve" (lines 197-198).

**R21 - Table 2, 3, 4,: Add to the captions that these Van Genuchten parameter values were fitted. For example, you could change 'representative of' to 'fitted to'. Furthermore, the symbols should contain units or an indication of no unit (e.g. [-]).**

A21 – We inserted the suggested modifications in Tables 2, 3 (now Table 4) and4 (now Table 5).

**R22 - Line 142-143: ", typical of the Mediterranean climate, characterized by warm and dry summer." can be removed. The climate the monitoring site has already been outlined in Section 2.**

A22 – We eliminated these information (lines 270-271).

**R23 - Line 145: What is the location of this local weather station? Could this be seen or shown in Fig. 3a?**

A23 – These temperature data are provided at a hourly scale since by the Pietrastornina weather station (located at 495 m a.s.l. and 15 km from Cervinara) and managed by the "Functional Centre for forecast, prevention and monitoring of risks and alerting for civil protection" of Campania Civil Protection Agency. We added this information (lines 235-237).

**R24 - Line 144-145: If 'Tmin', 'Tmed' and 'Tmax' are not used at other locations in the manuscript, there is no need to define them in the text here. Furthermore, the symbol 'Tmed' is confusing. Could be changed to 'Tmean' or 'Tave'? Are 'minimum' and 'maximum' also 'daily minimum' and 'daily maximum' like the 'daily mean'?**

A24 - Tmin, Tmed and Tmax (shown in Figure 8) refer to average monthly values of daily minimum, daily mean and daily maximum temperature. Such terms were left just in that figure; in particular "Tmed" was replaced by "Tmean".

**R25 - Line 154-155: Could the potential evapotranspiration be estimated specifically for this year?**

A25 - Unfortunately, we have no specific meteorological data available for the monitoring period at the monitoring site. A complete meteorological station has been installed at Cervinara in November 2017. You can find some recent data of potential evapotranspiration, estimated based on the acquired meteorological variables, in the cited reference Marino et al. (2020), so to have an idea of how current values can differ from the mean ones.

**R26 - Line 155-157: Is this relevant for the analyses to come?**

A26 –The reported information give an idea about the duration of the driest time window (essentially from May to October) that, as it will be shown in the following, is featured by the steepest paths in the water retention plane (Figures 10 and 12).

**R27 - Line 182-185: The part "…, which departs from the gravitational -1 only during rainfall." is not clear to me. Besides, the sentence is too long (too many elements).**

A27– We observed that in winter, when it does not rain, the vertical water potential gradient approaches "-1" (this happens when the vertical suction and/or water content profile tends to become constant), indicating that water (downward directed) flow is driven by gravity alone. Only during rainfall events the vertical water potential gradient departs from "-1" (this happens when the upper part of the soil profile becomes wetter owing to rain), indicating that capillarity favours the rapid rainwater infiltration.

The sentence was simplified (lines 321-323).

**R28 - Line 185-187: This sentence is not clear to me. What do you mean by "… the soil cover is being crossed by an intense downward flux …"?**

A28 - What we mean is that when periods of intense and frequent precipitations (and very small evapotranspiration, as it happens in winter) are associated with (nearly) constant soil water content and suction, this implies that the infiltrating water passes through the monitored soil layer without being retained (or being retained only in a little part). Hence, what we observe at z = 0.6 m and z = 1.0 m is the result of a downward water flow crossing the layer. When we see increasing water content, the infiltration from above is exceeding the drainage towards below, while the opposite occurs when we see decreasing water content.

We expanded this sentence in the revised manuscript to clarify this point (lines 323-327).

**R29 - Figure 8: Where does the hypothesized upper boundary come from?**

A29 – As explained in reply A7, the hypothesised upper boundaries have been expressed through the van Genuchten equation (1) assigning saturated and residual volumetric water contents ($\theta s = 0.75$; $\theta r = 0$) as for the main wetting curve (as shown by Table 4 and 5). Assuming $m = (n-1)/n$, according to Mualem (1976), the remaining $\alpha$ and n parameters have been chosen in order to provide the best fitting of the experimental points monitored during the time windows featured by the highest evapotranspiration effects.

**R30 - Line 189-191: Could you also report the parameters of these curves?**

A30–The plotted curve is just a manual fit. We remarked that in the modified text (line 329).

**R31 - Line 209-211: Change 'interpolated' to 'fitted'.**

A31–The term "interpolated" was replaced by "fitted" (line 352).

**R32 - Caption Table 3, Caption Table 4 and Line 205-213: I think all the time windows should be BC? Please check if it is correct that there is also referred to BCD and CD here.**

A32–From May, 8$^{th}$ (point B) to June, 6$^{th}$ (i.e. before June, 22$^{nd}$, that is representative of point C), we observed that at the depth z = 1.0 m the experimental points are effectively again well fitted by the previous path AB. After that date, instead, the points are well fitted by the upper boundary at the depth z=1.0 m too.

We preferred to avoid inserting between B and C another letter (not necessary to understand the overall response) in order not to induce confusion.

**R33 - Line 211-212: This is obvious; you already described that the paths are different.**

A33– Such redundant sentence was eliminated (line 354).

**R34 - Line 225-226: Change 'such a' to 'this'. Several similar occasions throughout the manuscript.**

A34– The suggested modification was reported there (line 369) and elsewhere.

**R35 - Line 226-227: Change 'is largely exceeding' to 'largely exceeds'. Add a comma after 'rainwater'. Remove 'owing to the action of plant roots' (you mean transpiration here, right?).**

A35– The suggested modifications will be reported (lines 371-372). Moreover, we confirm that "owing to the action of plant roots" refers to transpiration.

**R36 - Line 228-231: Perhaps these two short paragraphs can be combined into one. No need to add tables with fitted Van Genuchten parameters here?**

A36 –The two paragraphs were combined into one (line 375). It's not necessary to add tables, because the described experimental points are fitted by the van Genuchten curves, whose parameters have been already reported in Tables 4 and 5.

**R37 - Line 229: Is 'path BC' correct here?**

A37–Yes, "path BC" is correct. Points B, C and D belong to the same curve BCD.

**R38 - Line 255-256: What do you mean with "… did not accommodate the small evapotranspiration demand favouring an essentially downward flow …"?**

A38 - When the leaves of deciduous trees fall, the vegetation enters a dormant phase, during which they need very little water. Hence, even a small atmospheric evapotranspiration demand (i.e. the small estimated PET of winter months) is likely larger than the amount of water actually extracted from the soil by the vegetation. This is what we mean when we write that "vegetation probably did not accommodate the small evapotranspiration demand". We modified the sentence in the revised manuscript to clarify this point (lines 409-412).

**R39 - Line 266: Do you mean in the 48 hours before 5 December 2011?**

A39 – Yes, we mean that. The sentence was changed (lines 424-425).

**R40 - Line 273-275: One paragraph?**

A40 - The two paragraphs were combined into one (line 432).

**R41 - Line 278-285: One paragraph?**

A41 - The two paragraphs were combined into one (line 439).

**R42 - Line 300: Could this sentence be added to the paragraph above or below? Add a comma before 'suggesting'.**

A42 - This sentence was eliminated in the updated text (line 464).

---

## Author Response (AR2)

**COMMENTS TO THE AUTHOR**
**Few minor improvements should be addressed before final evaluation.**

**Reply by Authors**
Dear Topical Editor,
we took into account all the minor improvements suggested by Referee #3.
Please, also consider that:
- we added the copyright symbol © Google Earth to Figure 3, as requested by the "remarks from the precedent review file validation";
- we added in the "Acknowledgements" that the research has been developed with the support of the project VALERE 2019 funded by the Università degli Studi della Campania "Luigi Vanvitelli".

Best regards.
Luca  Comegna, on behalf of all the authors.

**SUGGESTIONS FOR REVISION OR REASONS FOR REJECTION (WILL BE PUBLISHED IF THE PAPER IS ACCEPTED FOR FINAL PUBLICATION)**

**OVERVIEW COMMENTS BY REFEREE #3**
**The authors have documented revisions to items pointed out by all reviewers and as such the manuscript is greatly improved. There are a few minor editorial items to be addressed.**

**Reply by Authors**
Dear Referee #3,

we really appreciated your positive assessment. The replies to the specific comments are reported in the follow. Please, consider that all the modifications will be referred to the lines of the marked-up version of the manuscript.

**LINE-BY-LINE COMMENTARY**

**R1 - In particular, some of the figure/table captions are minimal and not particularly informative. I suggest the authors revisit those and ask themselves if the reader can interpret the figures without the main body text. For example, captions for figures 1-3 are quite limited.**

A1- We added further information in the captions of all the Figures in order to allow their interpretation without the main body.

**R2 - Second, some of the sensors appear missing from table 3 (for example, where are the soil probes).**

A2 – Table 3 reports the main features of the electronic instruments (we added this clarification in the corresponding caption). The TDR probe simply consists of three 40 cm long metallic rods, having a diameter of 3 mm and spacing of 15 mm (it is shown by Fig. 7d and described at lines 189-191).

**R3 - Also, some of the error bounds are missing in the table for the various instruments.**

A3 - You are right. We added in Table 3 the accuracy of the rain gauge and of the TDR).

Regarding the accuracy of the Multiplexer, that is a connector (and not a measuring instrument), we added the expression "not defined" in Table 3.

**OTHER MINOR EDITORIALS**

**R4 - At line 73: What is meant by "five victims"? Be more specific...there were five human casualties or deaths?**

A4 - Unfortunately, there were five human deaths. We clarified that in text at line 82.

**R5 - At lines 95-105: Some of the sentences here in the text additions are too long and need to broken into similar text for reading ease.**

A5 – The sentences were broken in order to facilitate the reading (lines 108-117).

**R6 - There are other very minor editorials throughout, but those will addressed by copy editors if accepted.**

A6 – Of course, we will took into account further requested corrections that could be addressed by the copy editors.